# Diagnostic Significance of Influenza Symptoms and Signs, and Their Variation by Type/Subtype, in Outpatients Aged ≥ 15 Years: Novi Sad, Serbia

**DOI:** 10.3390/v17020272

**Published:** 2025-02-16

**Authors:** Ana Miljković, Aleksandra Patić, Vladimir Petrović, Nataša Nikolić, Gordana Kovačević, Tatjana Pustahija, Mioljub Ristić

**Affiliations:** 1Health Centre of Novi Sad, 21000 Novi Sad, Serbia; ana.miljkovic@mf.uns.ac.rs; 2Department of General Medicine and Geriatrics, Faculty of Medicine, University of Novi Sad, 21000 Novi Sad, Serbia; 3Institute of Public Health of Vojvodina, 21000 Novi Sad, Serbia; aleksandra.patic@mf.uns.ac.rs (A.P.); vladimir.petrovic@izjzv.org.rs (V.P.); natasa.nikolic@mf.uns.ac.rs (N.N.); tatjana.pustahija@mf.uns.ac.rs (T.P.); 4Department of Microbiology with Parasitology and Immunology, Faculty of Medicine, University of Novi Sad, 21000 Novi Sad, Serbia; 5Department of Epidemiology, Faculty of Medicine, University of Novi Sad, 21000 Novi Sad, Serbia

**Keywords:** influenza, surveillance system, case definition, symptoms, signs, primary healthcare system

## Abstract

This study assessed the diagnostic performance of influenza-related symptoms and signs and their combinations, as well as differences in patient characteristics based on the type/subtype of influenza, in outpatients at a primary healthcare surveillance system. Our prospective analysis included cases aged ≥ 15 years from two influenza seasons (2022/23 and 2023/24) in Novi Sad, Serbia. Influenza cases were confirmed using polymerase chain reaction (PCR) testing. The mean age of participants with laboratory-confirmed influenza was significantly lower than that of those without influenza (*p* < 0.0001): 37.90 vs. 54.92 years in 2022/23, and 40.21 vs. 54.17 years in 2023/24. Among the examined symptoms and signs, the highest sensitivity in the 2022/23 season was demonstrated for fever (87.95%, CI: 78.96–94.07), while in the 2023/24 season it was cough (100.00%, CI: 88.06–100.00). In the 2022/23 season, the positive predictive values (PPVs) were highest for fever (34.93%), chills (31.95%), myalgia (30.30%), and malaise (28.57%), but they dropped significantly in 2023/24 for all observed symptoms and signs (ranging from 1.91% to 9.17%). Compared to the World Health Organization’s case definition for influenza-like illness (ILI), the case definition provided by the European Centre for Disease Prevention and Control demonstrated higher sensitivity but lower specificity across both seasons. Participants who tested positive between December and February were more likely to have influenza A(H1N1)pdm09 or A(H3N2), whereas those who tested positive between February and April were more likely to have influenza B. This study underscores the importance of seasonal timing, symptom evaluation, and case definitions in improving influenza diagnosis in primary care.

## 1. Introduction

Influenza is one of the most studied infectious diseases, but significant gaps remain in understanding its epidemiology and developing effective prevention strategies [1,2]. While influenza A and B both contribute to seasonal outbreaks, only influenza A has caused pandemics, with six major events documented [3]. Globally, influenza affects 5–15% of the population annually, causing approximately 1 billion cases, 3–5 million severe cases, and 290,000–650,000 deaths, with Europe reporting about 150,000 fatalities [1,4,5]. The influenza virus exhibits significant antigenic variation, particularly antigenic drift, which presents challenges for public health systems. Its tendency to cause severe complications in high-risk groups—such as older adults, unvaccinated individuals, and those with comorbidities—amplifies its impact. Although primary influenza pneumonia is rare, it has a high fatality rate and exacerbates chronic respiratory diseases, including asthma and chronic obstructive pulmonary disease (COPD), as well as causing cardiovascular issues like myocarditis. Influenza-related mortality is about 1 per 1000 cases, with the highest rates in individuals aged ≥ 65 and children under 2 years [5]. These factors highlight the need for continued research in vaccine development, antiviral therapies, and public health interventions [1,2,3,4,5]. Influenza poses significant challenges due to its potential for seasonal epidemics and unpredictable outcomes, affecting both at-risk and previously healthy individuals [5,6]. Effective surveillance requires a comprehensive system to monitor trends, assess impact across demographics, and evaluate preventive interventions. Laboratory confirmation, with polymerase chain reaction (PCR) as the gold standard, is essential for accurate case identification. Standardized case definitions of influenza integrating clinical, epidemiological, and laboratory criteria ensure consistency, facilitate comparisons, and improve public health responses [1,3,6]. Surveillance systems must have high sensitivity, specificity, and positive predictive value to accurately identify true cases while minimizing false positives, particularly in distinguishing influenza-like illness (ILI) from acute respiratory infections (ARIs) [5,6,7]. During seasonal epidemics, routine laboratory testing for all suspected cases of influenza is often impractical, making validated ILI case definitions crucial [1,3,5,6,7]. Studies have evaluated the diagnostic validity of clinical features of influenza, focusing more on pediatric populations, but inconsistent findings have resulted from varying case definitions, surveillance timeframes, and laboratory resources. There are limited data on how pre-existing health status, vaccination history, or comorbid conditions can affect diagnostic accuracy [8,9,10,11]. The World Health Organization (WHO) and the European Centre for Disease Prevention and Control (ECDC) provide widely adopted ILI case definitions, but these often have low specificity, leading to false positives, especially during seasonal peaks of other respiratory viruses, such as respiratory syncytial virus or rhinoviruses [12,13]. Improving the specificity and accuracy of symptom-based case definitions is a global public health priority [3,7,9]. While the current ILI definitions are universal, their applicability across demographics, surveillance timelines, and clinical settings remains uncertain [14,15,16,17,18,19,20,21]. Only one study in our country has evaluated the diagnostic accuracy of the WHO’s ILI case definition, conducted in Vojvodina (Serbia). This study included ambulatory and hospitalized patients of all ages but did not consider pre-existing conditions, vaccination status, or contact history. The findings showed limited diagnostic utility, especially in older adults, with low specificity and a high rate of false positives. The study also did not assess whether specific symptoms were more predictive of influenza type A or B, or varied by viral subtype [22]. The objective of this study was to evaluate the diagnostic performance of symptoms and signs (symptoms/signs) associated with influenza both individually and in combination, as well as differences in outpatient characteristics based on the type/subtype of influenza, at the primary healthcare level, where initial medical evaluations typically occur.

## 2. Materials and Methods

### 2.1. Study Design

This study was designed as an observational, prospective, clinical–epidemiological investigation to identify predictors (symptoms/signs) of influenza. The study included three age groups of outpatients (15–29 years, 30–64 years, and ≥65 years) in the Health Centre of Novi Sad, Serbia.

To ensure a representative sample, this study adhered to a predefined minimum sample size for each age group in the city of Novi Sad (calculated using URL: http://www.raosoft.com/samplesize.html. (accessed on 31 May 2023)). During the two influenza surveillance seasons (2022/23 and 2023/24), spanning December to May (five months per season), the study aimed to enroll at least 18 patients per month from the 15–29 age group (a minimum of 90 patients per season and 180 across both seasons), 54 patients per month from the 30–64 age group (a minimum of 270 patients per season and 540 across both seasons), and 52 patients per month from the ≥65 age group (a minimum of 260 patients per season and 520 across both seasons). Overall, we planned to include a minimum of 1240 patients with suspected influenza over the 10-month surveillance period (620 patients per season).

In the Republic of Serbia, general practitioners (GPs) provide care for patients aged 15 years and older. The enrollment of patients was conducted by five GPs at the Health Centre of Novi Sad. Considering the minimum sample size required for this study, out of the 15 GPs previously involved in sentinel influenza surveillance, 5 expressed willingness to participate. The patient population under their care was sufficient to ensure a representative sample.

The inclusion criteria for patients in the study were as follows: patients aged 15 years and older, regardless of gender, presenting with one or more symptoms/signs—such as fever (including feverishness), cough, acute or gradual onset of illness, headache, dizziness, sore throat, rhinorrhea, myalgia, malaise, chills, loss of appetite, abdominal pain, nausea, vomiting, diarrhea, shortness of breath, or auscultatory findings suggestive of pneumonia—during the two influenza surveillance seasons, and consulting GPs during their first visit, were included in the study.

The exclusion criteria for patients were as follows: patients under 15 years of age, those without at least one symptom or sign related to influenza during the surveillance periods (prior to December or after the beginning of May in the subsequent year), and patients who visited GPs more than once were excluded from the study.

### 2.2. Study Components

#### 2.2.1. Clinical Examination

The patients were examined according to an established protocol, which included the measurement of height and weight for calculating the Body Mass Index (BMI), along with a review of medical records to determine the presence or absence of comorbidities. These data were recorded in a specifically designed questionnaire.

#### 2.2.2. Patient Survey

The patients were interviewed by a physician to complete the remaining sections of the questionnaire, which collected both clinical and epidemiological data pertinent to the potential transmission of the influenza virus.

Clinical data included the presence or absence of the following medical conditions:Hypertension;Cardiovascular diseases: myocardial infarction, heart failure, angina pectoris, arrhythmias, stroke (cerebrovascular accident);Asthma;Diabetes mellitus;Obesity (BMI ≥ 30 classified as obese).

Along with information on age, gender, employment status, and marital status, the questionnaire also included data on symptoms/signs associated with influenza, smoking habits, frequency of alcohol consumption, use of public or taxi transportation, prior hospitalization due to influenza, and the month of inclusion in the study. Additionally, it collected epidemiological data such as influenza vaccination status, close contact with similarly affected individuals, and living with school-aged children within the household.

### 2.3. Laboratory Examination

The training of all included GPs in adequate sampling and the sample-handling procedures was conducted before the start of research and lasted for 10 days. Nasopharyngeal swabs were collected for PCR testing to confirm influenza infection. The samples were analyzed in the Centre for Virology at the Institute of Public Health of Vojvodina, Novi Sad.

### 2.4. Ethical Considerations

The study protocol was approved by the Ethics Committees of the Faculty of Medicine, University of Novi Sad, and the Health Centre of Novi Sad, Novi Sad, Serbia. The respective approval reference numbers are as follows: Faculty of Medicine, University of Novi Sad: 01-39/164/1 (1 February 2022); Health Centre of Novi Sad: 21/3-1 (10 February 2022). Participation in this study was voluntary, anonymous, and without material compensation. Written informed consent for this study was obtained from all participants at the time of sampling, which took place during the study period. Each respondent was assigned with a unique code, after which all data were entered into a database on a computer with protected access limited to members of the research team.

### 2.5. Statistical Analysis

Statistical analyses were performed using the Statistical Package for Social Sciences (SPSS) version 21. Numerical data were expressed as means (arithmetic mean, median) and measures of variability (interquartile range, standard deviation), while categorical data were presented as frequencies and percentages. For the comparison of continuous variables, Student’s *t*-test was used. Univariate linear regression analysis (odds ratio [OR] with 95% confidence intervals [CIs]) was conducted to examine potential predictors of laboratory-confirmed influenza, with the lowest percentages of PCR-confirmed influenza stratified by observed characteristics used as the reference category. The validity of case definitions and various symptom/sign combinations, including sensitivity, specificity, positive predictive value (PPV), and negative predictive value (NPV), was calculated. The positive likelihood ratio (LR+) and negative likelihood ratio (LR−), along with their 95% CIs, were also determined, representing the ratio of the probability of a combination of symptoms/signs being present among individuals with laboratory-confirmed influenza compared to those with a negative influenza test. Additionally, the accuracy of the test and Cohen’s kappa coefficient (κ) with 95% CIs were calculated for each symptom/sign and their various combinations.

Based on these results, the diagnostic significance of the WHO and ECDC ILI case definitions, as well as other symptoms/signs and their combinations, was assessed to determine which had the highest likelihood of yielding a positive PCR test for influenza across two consecutive surveillance seasons. Tests of proportion were conducted to compare characteristics between participants with three types/subtypes of influenza virus (A(H1N) pdm09, A(H3N2), and B).

Values of OR > 1.1 and LR+ ≥ 2 for certain symptom and sign combinations were considered to be thresholds indicating an increased probability of laboratory-confirmed influenza. A test with an accuracy value above 0.710 was deemed to have useful diagnostic value [23]. Statistical significance was defined as a *p*-value < 0.05.

## 3. Results

### 3.1. Characteristics of Participants Included in the Study During the 2022/23 Influenza Season

A total of 679 participants were included in the study, mostly female (442/679; 65.1%). Out of the total number of samples collected (679), 83 (12.2%) tested positive for influenza. Various sociodemographic, clinical, epidemiological, and behavioral characteristics were significantly associated with laboratory-confirmed influenza (Table 1).

Participants aged 15–29 were significantly more likely to have laboratory-confirmed influenza compared to those ≥65 years old (OR = 12.29; 95% CI: 5.18–29.15; *p* < 0.0001). Similarly, participants aged 30–64 were also at higher risk compared to the ≥65 age group (OR = 6.97; 95% CI: 3.09–15.68; *p* < 0.0001). In addition, the mean age of participants with influenza was significantly lower (37.90 years) compared to those who tested negative (54.92 years) (*p* < 0.0001). Being employed (OR = 7.27; 95% CI: 3.22–16.45; *p* < 0.0001) or unemployed (OR = 10.39; 95% CI: 4.44–24.30; *p* < 0.0001) was strongly associated with laboratory-confirmed influenza compared to retired individuals. Unmarried participants were significantly more likely to have influenza compared to widowed participants (OR = 6.18; 95% CI: 1.85–20.61; *p* = 0.0031).

Regarding observed symptoms/signs, strong associations with laboratory-confirmed influenza were reported for fever (OR = 24.69; 95% CI: 12.41–49.13; *p* < 0.0001), cough (OR = 5.35; 95% CI: 2.84–10.10; *p* < 0.0001), sudden onset of symptoms (OR = 7.96; 95% CI: 4.59–13.80; *p* < 0.0001), malaise (OR = 12.95; 95% CI: 6.99–24.02; *p* < 0.0001), myalgia (OR = 8.66; 95% CI: 5.16–14.51; *p* < 0.0001), and chills (OR = 7.79; 95% CI: 4.75–12.78; *p* < 0.0001). Moderately significant symptoms/signs included headache (OR = 3.90; *p* < 0.0001), dizziness (OR = 2.44; *p* = 0.0011), loss of appetite (OR = 2.78; *p* = 0.0004), and vomiting (OR = 3.14; *p* = 0.0140). Sore throat was significantly associated with a negative result for influenza (OR = 0.40; 95% CI: 0.22–0.71; *p* = 0.0020). Participants without influenza were more likely to have a sore throat compared to those with laboratory-confirmed influenza (90.10% vs. 78.31%, OR = 0.40, 95% CI: 0.22–0.71, *p* = 0.0020). Those with laboratory-confirmed influenza were significantly less likely to have been vaccinated against the seasonal influenza compared to those who tested negative (OR = 0.36; 95% CI: 0.15–0.85; *p* = 0.0195). A similar trend was observed for vaccination against influenza in the previous year (OR = 0.31; 95% CI: 0.11–0.88; *p* = 0.0272). Those with influenza were also less likely to have received the influenza vaccine in the current season, although the association was weaker (OR = 0.39; 95% CI: 0.15–0.99; *p* = 0.0484). Participants who tested negative for influenza were more likely to have been vaccinated against COVID-19 in a timely manner (OR = 0.42; 95% CI: 0.26–0.67; *p* = 0.0002).

Participants with influenza were significantly less likely to have hypertension compared to those without influenza (OR = 0.28; 95% CI: 0.16–0.48; *p* < 0.0001). A significantly higher proportion of participants without influenza had other chronic diseases (46.48%) compared to those with influenza (28.92%) (OR = 0.47; 95% CI: 0.28–0.77; *p* = 0.0030). A significant association was found between the absence of chronic diseases and testing positive for influenza (OR = 3.30; 95% CI: 2.06–5.27; *p* < 0.0001). Participants who had contact with someone who had influenza -like symptoms seven days prior to testing were significantly more likely to test positive for influenza (OR = 5.02; 95% CI: 2.84–8.87; *p* < 0.0001). There was only one previous hospitalization due to confirmed influenza. Participants included in the study from 1 December 2022 to 14 February 2023 were more likely to test positive for influenza compared to those included between 15 February 2023 and 30 April 2023 (OR = 1.63; 95% CI: 1.02–2.59; *p* = 0.0408).

### 3.2. Characteristics of Participants Included in the Study During the 2023/24 Influenza Season

A total of 755 participants were included in the study, mostly female (475/755; 62.91%). Out of the total number of samples collected (755), 29 (3.8%) tested positive for influenza. Different sociodemographic, clinical, epidemiological, and behavioral characteristics were significantly associated with laboratory-confirmed influenza (Table 2).

Participants aged 15–29 were significantly more likely to have laboratory-confirmed influenza compared to those ≥65 years old (OR = 9.96; 95% CI: 3.17–31.30; *p* < 0.0001). In addition, participants with confirmed influenza were significantly (*p* < 0.0001) younger on average (mean age 40.21 years), compared to those who were influenza-negative (mean age 54.17 years). Being employed (OR = 3.78; 95% CI: 1.24–11.52; *p* = 0.0195) or unemployed (OR = 4.30; 95% CI: 1.33–13.93; *p* = 0.0150) was strongly associated with laboratory-confirmed influenza compared to retired individuals. Based on observed symptoms/signs, a strong association with laboratory-confirmed influenza was recorded for fever (OR = 22.27; 95% CI: 5.25–94.39; *p* < 0.0001), sudden onset of symptoms (OR = 4.28; 95% CI: 1.92–9.55; *p* = 0.0004), headache (OR = 7.32; 95% CI: 3.08–17.40; *p* < 0.0001), and malaise (OR = 4.03; 95% CI: 1.62–10.01; *p* = 0.0027). Nasal congestion (OR = 0.20; 95% CI: 0.09–0.43; *p* < 0.0001) was less common in individuals with influenza compared to those without laboratory-confirmed influenza.

There was a lower prevalence of hypertension among influenza-positive participants (24.14%) than in those who tested negative (46.28%), with a significant inverse association (OR = 0.37; 95% CI: 0.16–0.88; *p* = 0.0237). A large majority of participants were non-smokers (90.60% overall; 86.21% among influenza-positive cases). Smokers with a history of ≤5 years exhibited a borderline statistically significant higher prevalence among influenza-positive participants compared to influenza-negative participants (OR = 24.09; 95% CI: 1.01–574.85, *p* = 0.0493). This finding suggests a potential link, although the wide confidence interval indicates limited precision.

There were no reported previous hospitalizations due to confirmed influenza. Participants included in the study from 1 December 2023 to 14 February 2024 were more likely to test positive for influenza compared to those included between 15 February 2024 and 30 April 2024 (OR = 4.04; 95% CI: 1.39–11.73, *p* = 0.0103).

### 3.3. Diagnostic Utility of Symptoms/Signs of Influenza in the 2022/23 and 2023/24 Influenza Seasons

We evaluated the diagnostic utility of various symptoms/signs, characterized by their sensitivity (Se), specificity (Sp), positive predictive value (PPV), negative predictive value (NPV), likelihood ratios (LR+ and LR−), accuracy, and inter-rater reliability (kappa), during two influenza seasons.

Among the examined symptoms/signs, fever had the highest sensitivity (87.95%, CI: 78.96–94.07) in the 2022/23 season, and cough was the most sensitive (100.00%, 95% CI: 88.06–100.00) in the 2023/24 season, for laboratory-confirmed influenza. But, due to the low specificity of cough in the 2023/24 season (13.91%, 95% CI: 11.48–16.64), the accuracy was higher for fever in both seasons (0.785, CI: 0.763–0.799, and 0.634, 95% CI: 0.621–0.639, respectively). Except for fever, high sensitivity and good balance between sensitivity and specificity (LR+ ≥ 2 and accuracy > 0.710) were found in the 2022/23 season for myalgia (LR+ 3.12, accuracy = 0.763), malaise (LR+ 2.87, accuracy = 0.723), chills (LR+ 3.37, accuracy = 0.788), loss of appetite (LR+ 2.35, accuracy = 0.817), and vomiting (LR+ 2.96, accuracy = 0.863). The highest specificity in this season was observed for the following gastrointestinal symptoms/signs: vomiting (97.15%, CI: 95.47–98.33), diarrhea (95.81%, CI: 93.87–97.27), nausea (93.62%, CI: 91.35–95.45), abdominal pain (92.79%, CI: 90.40–94.73), and loss of appetite (89.77%, CI: 87.05–92.08), indicating their value in ruling out influenza during this season.

In the 2023/24 season, the highest values of sensitivity, LR+, and accuracy were found for headache (Se = 75.86%, LR+ 2.53, accuracy = 0.702). Similar to the 2022/23 season, the highest values of specificity in the 2023/24 influenza season were observed for gastrointestinal symptoms/signs (loss of appetite, abdominal pain, nausea, vomiting, and diarrhea), ranging from 88.15% to 98.21%. The PPV in the 2022/23 season was the highest for fever (34.93%, CI: 31.22–38.83), chills (31.95%, CI: 27.22–37.09), myalgia (30.30%, CI: 26.29–34.64), and malaise (28.57%, CI: 25.51–31.84), while the PPV in the following season had lower values for the observed symptoms/signs, ranging from 1.91% to 9.17% (Table 3).

The analysis of symptoms/signs and other variables during the 2022/23 season, stratified by age groups, is detailed in Appendix A. Similarly, the diagnostic utility of selected signs and symptoms for detecting influenza and other variables during the 2023/24 season, categorized into three age groups, is outlined in Appendix A.

### 3.4. Performance of Symptoms/Signs Related to Influenza and Case Definitions of ILI During the Two Seasons (2022/23 and 2023/24), by Age Group

The diagnostic value of the selected symptom/sign combinations for influenza, stratified in three age groups and throughout two seasons, is shown in Table 4.

#### 3.4.1. WHO ILI Case Definition: Fever (≥38 °C) Within 10 Days + Cough Without Sudden Onset of Symptoms/Signs

In the 2022/23 season, sensitivity for this combination of symptoms/signs was consistently low across all age groups, ranging from 10.42% (95% CI: 3.47–22.66) in the 30–64 age group to 14.29% (95% CI: 0.36–57.87) in the ≥65 age group. Specificity was high across all groups, with the highest value of 96.46% (95% CI: 93.38–98.37) observed in participants aged 30–64 years. Accuracy was as follows: 0.750 (95% CI: 0.712–0.788) in the 15–29 age group, 0.828 (95% CI: 0.808–0.853) in the 30–64 age group, and 0.936 (95% CI: 0.929–0.957) in the ≥65 age group, with a total accuracy of 0.857 (95% CI: 0.844–0.874).

In the 2023/24 season, sensitivity for this combination of symptoms/signs was highest in the youngest age group (46.15%, 95% CI: 19.22–74.87) but significantly lower in the middle-aged (8.33%, 95% CI: 0.21–38.48) and ≥65 age groups (25.00%, 95% CI: 0.63–80.59). Specificity ranged from 78.26% (15–29 years) to 92.55% (≥65 years). The overall accuracy for this combination of symptoms/signs was 0.848% (95% CI: 0.837–0.862).

#### 3.4.2. WHO ILI Case Definition Plus Sudden Onset of Symptoms/Signs

In the 2022/23 season, this combination of symptoms/signs yielded higher sensitivity, especially in the 15–29 age group at 78.57% (95% CI: 59.05–91.70). Specificity ranged from 64.29% (95% CI: 53.08–74.45) in the youngest age group to 93.41% (95% CI: 89.66–96.12) in the ≥65 age group. Accuracy was 0.679 (95% CI: 0.591–0.738) in the 15–29 age group, 0.795 (95% CI: 0.752–0.833) in the 30–64 age group, and 0.925 (95% CI: 0.905–0.941) in the ≥65 age group, with a total accuracy of 0.826 (95% CI: 0.801–0.848). In the 2023/24 season, sensitivity for this combination of symptoms/signs was markedly high in the 30–64 age group (91.67%, 95% CI: 61.52–99.79), while it remained moderate in the younger group (46.15%, 95% CI: 19.22–74.87) and the ≥65 age group (50.00%, 95% CI: 6.76–93.24). Specificity for this combination of symptoms/signs varied between 72.83% and 90.07% by age group, and the overall accuracy was 0.803 (95% CI: 0.788–0.815).

#### 3.4.3. ECDC ILI Case Definition: Sudden Onset and at Least One Among Fever, Feverishness, Headache, Malaise, and Myalgia, and at Least One Among Cough, Sore Throat, and Shortness of Breath

In the 2022/23 season, this combination of symptoms/signs demonstrated high sensitivity, particularly in the 15–29 age group (85.71%; 95% CI: 67.33–95.97). However, due to low specificity in this age group, the overall accuracy was higher among patients aged 30–64 and ≥65 (0.709 and 0.879, respectively, compared to 0.545). Specificity was the lowest observed among all combinations of symptoms/signs, ranging from 44.05% (95% CI: 33.22–55.30) in the youngest age group to 88.76% (95% CI: 84.26–92.34) in the ≥65 age group. In the 2023/24 season, sensitivity for this combination of symptoms/signs was markedly high in the 30–64 age group (91.67%, 95% CI: 61.52–99.79), while it remained moderate in the younger group (53.85%, 95% CI: 25.13–80.78) and the ≥65 age group (50.00%, 95% CI: 6.76–93.24). Specificity for this combination of symptoms/signs varied between 58.70% and 85.11% by age group, and the overall accuracy was 0.714 (95% CI: 0.699–0.725).

### 3.5. Combinations of Symptoms/Signs Across Age Groups

#### 3.5.1. Combination of Symptoms/Signs: Fever (Including Feverishness), Headache, Malaise, Myalgia + Cough

In the 2022/23 season, this combination of symptoms/signs exhibited moderate sensitivity, with the highest value of 57.14% (95% CI: 37.18–75.54) in the 15–29 age group. Specificity was robust across all groups, peaking at 97.67% (95% CI: 95.01–99.14) in the ≥65 age group. Accuracy was 0.768 (95% CI: 0.684–0.842) in the 15–29 age group, 0.825 (95% CI: 0.792–0.859) in the 30–64 age group, and 0.955 (95% CI: 0.948–0.974) in the ≥65 age group, with a total accuracy of 0.866 (95% CI: 0.845–0.887). In the 2023/24 season, sensitivity for this combination of symptoms/signs was low across all age groups, ranging from 0.00% (≥65 years) to 16.67% (30–64 years). Specificity was consistently high, reaching 98.23% (≥65 years). The total accuracy for this combination of symptoms/signs was 0.919 (95% CI: 0.912–0.932).

#### 3.5.2. Combination of Symptoms/Signs: Fever (Including Feverishness), Headache, Malaise, Myalgia + Sore Throat

In the 2022/23 season, sensitivity for this combination of symptoms/signs ranged from 14.29% (95% CI: 0.36–57.87) in the ≥65 age group to 57.14% (95% CI: 37.18–75.54) in the youngest age group. Specificity was consistently high, with a maximum of 98.45% (95% CI: 96.08–99.58) in the ≥65 age group. Accuracy was 0.786 (95% CI: 0.703–0.857) in the 15–29 age group, 0.818 (95% CI: 0.787–0.852) in the 30–64 age group, and 0.962 (95% CI: 0.955–0.980) in the ≥65 age group, with a total accuracy of 0.869 (95% CI: 0.848–0.890). In the 2023/24 season, sensitivity for this combination of symptoms/signs was very low, varying from 0.00% (≥65 years) to 8.33% (30–64 years), while specificity remained high (94.89–98.94%). The overall accuracy for this combination of symptoms/signs was 0.932 (95% CI: 0.928–0.944).

#### 3.5.3. Combination of Symptoms/Signs: Fever (Including Feverishness), Headache, Malaise, Myalgia + Shortness of Breath

In the 2022/23 season, this combination of symptoms/signs demonstrated the lowest sensitivity, ranging from 0.00% in the ≥65 age group to 10.71% (95% CI: 2.27–28.23) in the 15–29 age group. Specificity was consistently high, exceeding 98% in all age groups. Accuracy was 0.768 (95% CI: 0.730–0.785) in the 15–29 age group, 0.841 (95% CI: 0.824–0.858) in the 30–64 age group, and 0.962 (95% CI: 0.962–0.977) in the ≥65 age group, with a total accuracy of 0.876 (95% CI: 0.866–0.888). In the 2023/24 season, sensitivity for this combination of symptoms/signs was the lowest across all groups (0.00–8.33%), while specificity was the highest, exceeding 96% in all age groups. The overall accuracy for this combination of symptoms/signs was 0.946 (95% CI: 0.941–0.956) (Table 4).

### 3.6. Differences Between Types of Influenza Virus Regarding Characteristics of Participants in the 2022/23 and 2023/24 Influenza Seasons

The differences by type of influenza regarding certain variables in the 2022/23 and 2023/24 seasons are given in Appendix A, respectively. During the 2022/23 season, no statistically significant differences were observed between influenza virus types across the three age groups of participants (*p* > 0.05). Regarding symptoms/signs associated with influenza, participants with influenza A(H1N1)pdm09 were borderline more likely to experience a sudden onset of symptoms compared to those with influenza A(H3N2) (83.33% vs. 54.55%, *p* = 0.0509) and significantly more likely to report myalgia (80.56% vs. 45.45%, *p* = 0.0242). Conversely, participants with A(H1N1)pdm09 were less likely to report headaches compared to those with A(H3N2) (58.33% vs. 90.91%, *p* = 0.0483) and less likely to report malaise compared to participants with influenza B (77.78% vs. 94.44%, *p* = 0.0424).

Patients with influenza A(H3N2) were more likely to experience headaches (90.91% vs. 55.56%, *p* = 0.0346) and dizziness (54.55% vs. 22.22%, *p* = 0.0423) compared to those with influenza B. In contrast, participants with influenza B were more likely to report sore throat, malaise, and chills compared to those with influenza A(H3N2) (86.11% vs. 54.55%, *p* = 0.0268; 94.44% vs. 72.73%, *p* = 0.0432; and 77.78% vs. 45.45%, *p* = 0.0423, respectively). Additionally, participants with other medical conditions not complicating influenza were more likely to test positive for influenza A(H3N2) than A(H1N1)pdm09 or influenza B (63.64% vs. 27.78%, *p* = 0.0335; and 63.64% vs. 19.44%, *p* = 0.0055, respectively). Participants who smoked were more likely to test positive for influenza A(H1N1)pdm09 compared to influenza B (33.33% vs. 11.11%, *p* = 0.0243). Regarding the surveillance period, participants tested between December 2022 and February 2023 were significantly more likely to test positive for influenza A(H1N1)pdm09 compared to influenza B (77.78% vs. 38.89%, *p* = 0.0009). Conversely, cases observed between February and April 2023 were more likely to test positive for influenza B compared to A(H1N1)pdm09 (61.11% vs. 22.22%, *p* = 0.0009). Other comparisons between influenza types were not statistically significant.

During the 2023/24 season, participants aged 15–29 years were borderline significantly more likely to test positive for influenza A(H3N2) compared to influenza A(H1N1)pdm09 (57.89% vs. 14.29%, *p* = 0.0524). Regarding symptoms/signs associated with influenza, patients presenting with dizziness were significantly more likely to test positive for influenza A(H1N1)pdm09 compared to influenza A(H3N2) (28.57% vs. 0%, *p* = 0.0174). Participants without chronic diseases were borderline significantly more likely to test positive for influenza B compared to influenza A(H1N1)pdm09 (100% vs. 28.57%, *p* = 0.0495). Additionally, patients living with children aged 15–19 years were significantly more likely to test positive for influenza B compared to influenza A(H3N2) (33.33% vs. 0%, *p* = 0.0119). Participants tested between December 2023 and February 2024 were significantly more likely to test positive for influenza A(H1N1)pdm09 and influenza A(H3N2) compared to influenza B (85.71% vs. 0%, *p* = 0.0162, and 100% vs. 0%, *p* < 0.0001, respectively). Conversely, between February and April 2024, influenza B was detected significantly more frequently compared to influenza A(H1N1)pdm09 and influenza A(H3N2) (100% vs. 14.29%, *p* = 0.0162, and 100% vs. 0%, *p* < 0.0001, respectively.

## 4. Discussion

This study evaluated the symptoms/signs associated with laboratory-confirmed influenza and the performance of influenza case definitions as well as differences in patients’ characteristics regarding the type/subtype of influenza in outpatients aged ≥15 years, using data obtained from a primary healthcare influenza surveillance system. The epidemiological and virological characteristics of influenza were analyzed in a prospective study conducted during two consecutive influenza seasons (2022/23 and 2023/24) in Novi Sad, Serbia. All influenza types and subtypes, including A(H1N1)pdm09, A(H3N2), and influenza B, were identified.

Our findings reveal a strong association between younger age groups and laboratory-confirmed influenza. Participants aged 15–29 and 30–64 years in the 2022/23 season, as well as those aged 15–29 years in the 2023/24 season, showed significantly higher influenza positivity rates compared to individuals aged ≥65 years. This aligns with previous studies highlighting younger adults’ increased susceptibility due to greater social interaction and exposure risk [24]. While gender differences in influenza outcomes can be age- and country-dependent [25], our results suggest that gender is not a primary risk factor. Employment status, however, was significantly associated with influenza risk, with both employed and unemployed individuals more likely to test positive than retired individuals. This may reflect increased occupational exposure and reduced adoption of preventive measures among younger populations compared to retirees, who likely maintained practices from the COVID-19 pandemic, such as mask-wearing and physical distancing [25,26]. Typical manifestations of influenza include uncomplicated upper respiratory tract symptoms, with or without fever, and in some instances, progression to severe disease [27]. In our study, the symptoms/signs most strongly associated with influenza during the 2022/23 season were fever, cough, and sudden onset of symptoms. During the 2023/24 season, fever, sudden onset of symptoms, headache, and malaise were the most prominent. These findings align with previous research identifying these symptoms/signs as key indicators of influenza [27].

Although only a small proportion of the participants were vaccinated, individuals with laboratory-confirmed influenza during the 2022/23 season were significantly less likely to have received the seasonal influenza vaccine compared to those who tested negative. Additionally, influenza-negative participants were more likely to have received timely COVID-19 vaccination. The negative association between influenza positivity and timely COVID-19 vaccination may reflect competing vaccination priorities or differing health behavior patterns associated with these two diseases. It can be assumed that participants who were vaccinated against COVID-19 were also more likely to wear masks, maintain physical distance, and follow other preventive measures, which may have reduced the risk of influenza virus infection. In contrast, no significant association between vaccination status and laboratory-confirmed influenza was identified during the 2023/24 season, a period marked by the relaxation of all COVID-19 preventive measures.

Regarding comorbidities, we found that hypertension was more common in patients without influenza than in those with influenza in both seasons. In the 2022/23 season, patients without influenza were more likely to have chronic diseases that did not complicate influenza and were without comorbidities than those with laboratory-confirmed influenza. Diabetes mellitus type 2 showed a borderline association with laboratory-confirmed influenza. These observations may be attributed to the inclusion of participants solely at the primary care level, as well as the fact that the majority of participants were young adults.

During the 2022/23 influenza season, participants with recent contact with individuals showing influenza-like symptoms were more likely to test positive for influenza, reflecting the primary role of person-to-person transmission in close-contact settings. Household transmission studies indicate a transmission risk of up to 38% within households, with secondary cases typically appearing three days after the index case, and younger age has also been linked to increased susceptibility [28]. In our study, most participants did not live with school-aged children, and no association with laboratory-confirmed influenza was observed for this characteristic. During the 2023/24 season, smokers with a smoking history of ≤5 years were significantly more likely to test positive for influenza, suggesting a potential link between smoking and influenza susceptibility, despite limited precision. Smoking has been consistently identified as a major risk factor for respiratory and other infections [29]. In the 2022/23 season, participants aged ≥65 years who smoked had a fourfold higher likelihood of testing positive for influenza.

We found that influenza positivity was higher from December to February compared to February to April in both seasons, aligning with the typical winter peak of influenza outbreaks [30,31,32]. The diagnostic utility of symptoms/signs related to influenza differed between the 2022/23 and 2023/24 seasons, impacting clinical practice. Fever consistently emerged as a reliable indicator, with participants with fever being about four times (LR+ 3.85) more likely to test positive during 2022/23 and three times (LR+ 2.47) more likely during 2023/24. While fever’s high negative predictive value (above 97%) supports its role in ruling out influenza, its positive predictive value is limited due to its non-specificity. Previous studies confirm fever as a hallmark symptom, but up to 38% of influenza inpatients lacked fever at admission [33]. A Singapore study during the 2009 pandemic reported that fever had a sensitivity of 61%, specificity of 72%, PPV of 16%, and NPV of 95% [34]. Compared to our findings, fever showed higher sensitivity and NPV but lower specificity and PPV in 2023/24. Similarly, a Taiwanese study (2010–2012) reported that fever had a sensitivity, specificity, PPV, and NPV of 88%, 34%, 53%, and 79%, respectively [35], with our results showing higher sensitivity, specificity, and NPV.

Fever’s role as a predictor of influenza differs between children and adults. A Canadian study (2008–2011) found that fever was less common in adults than in children [16]. Our study, focused on adults, confirmed fever as the most reliable predictor of influenza, showing the highest diagnostic performance among the evaluated symptoms.

During the 2022/23 season, high sensitivity and diagnostic accuracy were noted for sudden symptom onset, myalgia, malaise, chills, loss of appetite, and vomiting, while in 2023/24 this applied only to cough and headache. Symptoms such as sore throat, nasal congestion, and gastrointestinal symptoms/signs showed limited diagnostic reliability, consistent with prior research [36,37]. In a previous study [35], the sensitivity for cough and nasal congestion was 97% and 66%, respectively, with PPVs of 52% and 57%, respectively. We observed higher sensitivity for cough in 2023/24 and nasal congestion in 2022/23, although the PPVs were lower. These findings underscore the variability in the presentation of symptoms/signs across seasons and age groups, complicating influenza diagnosis based on clinical signs alone. 

Among young adults (15–29 years), fever (≥38 °C) and cough were common in both seasons, with sudden onset, headache, and malaise specific to 2022/23. Gastrointestinal symptoms/signs like vomiting and diarrhea, though low in sensitivity, showed high specificity, making them useful for ruling out influenza. In middle-aged adults (30–64 years), fever, myalgia, and chills were key indicators in 2022/23, while fever, cough, headache, sudden onset, and malaise dominated in 2023/24. Gastrointestinal symptoms/signs also exhibited low sensitivity but high specificity in this group. A French study (2009–2014) found fever, cough, chills, and coryza/rhinitis associated with influenza in patients aged 15–64, while other symptoms showed no significant association [38]. In adults aged ≥ 65 years, fever, sudden onset, myalgia, malaise, and chills were reliable in 2022/23, with fever and sudden onset remaining prominent in 2023/24. Shortness of breath and pneumonia significantly increased the likelihood of influenza (LR+ 4.10 and LR+ 5.27, respectively) in 2022/23. The French study similarly highlighted fever, cough, weakness, and chills as reliable indicators in this age group, with other symptoms/signs showing less diagnostic accuracy.

Using the WHO ILI case definition (fever ≥ 38 °C within 10 days and cough, without sudden onset), we found that sensitivity was low across all age groups: 11% in 2022/23 and 28% in 2023/24. Adding sudden onset to fever and cough improved sensitivity but reduced specificity across all age groups in both seasons. Similarly, a multicenter study of 11,277 pregnant women (2017–2018) in India, Peru, and Thailand found that the WHO ILI case definition had low sensitivity (16%) but high specificity (98%) for laboratory-confirmed influenza. Expanding the definition to include subjective fever modestly increased sensitivity (25%), with a slight reduction in specificity (95%), while including subjective fever or chills further raised sensitivity (32%) but lowered specificity (91%) [9]. When testing the ECDC criteria (all symptoms together: fever with feverishness, headache, malaise, myalgia, and cough), sensitivity was moderate across all age groups in the 2022/23 season (36.14%) but lower in the 2023/24 season (13.79%). Similar sensitivity values were observed for the ECDC criteria along with sore throat, ranging from 33.73% (2022/23 season) to 6.90% (2023/24 season). The lowest sensitivity across both seasons was recorded for the ECDC criteria along with shortness of breath, with values ranging from 8.43% (2022/23 season) to 6.90% (2023/24 season). Interestingly, the last mentioned combination showed the highest specificity across both seasons, with values of 98.66% in 2022/23 and 98.07% in 2023/24. High specificity suggests that these combinations are stringent, effectively ruling out individuals who do not have influenza. However, the low sensitivity indicates that a significant proportion of true influenza cases are missed due to the stringent criteria. This could lead to under-diagnosis and inadequate management of influenza cases. In public health surveillance, these limitations may result in underestimating the true burden of influenza in the population. When comparing our findings with other available data, another study reported that the WHO ILI case definition had a sensitivity of 89.8%, specificity of 21.4%, and accuracy of 0.556 [38]. In contrast, our results over two influenza seasons showed sensitivity ranging from 10.84% to 27.59%, specificity from 87.05% to 96.14%, and accuracy from 0.848 to 0.857. Although the exact reasons for these discrepancies remain unclear, it is likely that a substantial proportion of our participants exhibited symptoms/signs overlapping with those of COVID-19, which persisted throughout the study period. Additionally, the study [38] included data from the 2009/10 influenza pandemic, during which individuals presenting with specific respiratory symptoms were more likely to test positive for influenza. Notably, nearly all patients (96%) in the mentioned study had a fever ≥ 38 °C, whereas, in our study, fever was observed only in 31% of patients during the 2022/23 influenza season and in 40% of patients during the 2023/24 season. Similarly, the referenced study [38] reported that the ECDC criteria achieved a sensitivity of 96.1%, specificity of 6.6%, and accuracy of 0.513. Our findings suggest that including more symptoms/signs in diagnostic criteria may reduce sensitivity while increasing specificity. Conversely, using broader criteria (e.g., requiring at least one symptom) may improve sensitivity, but at the cost of lower specificity.

A study in Senegal (657 patients) evaluated the WHO, CDC, and ECDC ILI case definitions. Sensitivity (81.0%) and NPV (91.0%) were similar across all definitions, but the WHO and CDC criteria had the highest specificity (52.0%) and PPV (32.0%). Symptomatic predictors of influenza varied by age [39]. The higher sensitivity in the Senegalese study compared to ours likely reflects their inclusion of fever as a mandatory symptom, unlike in our study, where fever was less prevalent. The authors noted that this might have overestimated sensitivity and underestimated specificity. Notably, the mentioned study was conducted before the COVID-19 pandemic (2013–2016).

A study conducted in Catalonia, Spain, within a primary healthcare sentinel surveillance network over 11 years (2008–2018), analyzed 10,367 participants and assessed the sensitivity and specificity of the WHO and ECDC ILI case definitions [7]. The WHO case definition had a sensitivity of 82% and specificity of 37%, while the ECDC case definition had 58% sensitivity and 52% specificity. These values remained consistent across epidemic and non-epidemic periods. The WHO case definition performed best in individuals aged ≥ 65, similar to our findings in the 2022/23 season. In contrast, our study showed lower sensitivity but higher specificity for the WHO case definition. Combining it with sudden onset increased sensitivity sixfold in 2022/23 and doubled it in 2023/24, while maintaining high specificity. For the ECDC case definition during both seasons, our sensitivity (75.90% and 68.97%) and specificity (74.66% and 71.49%) were significantly higher, with accuracy values of 0.748 and 0.7142. In both studies, fever and cough had the highest sensitivity, but gastrointestinal symptoms/signs showed higher specificity than shortness of breath in our study. When comparing the results from the Catalonian study with our findings, differences between the studies could be due to the Catalonian study’s longer duration (11 years vs. 2 years), pre-COVID-19 vs. COVID-19 periods, and differences in laboratory-confirmed influenza prevalence (31.3% in Catalonia vs. 12.2% in 2022/23 and 3.8% in 2023/24). Additionally, 52% of the Catalonian participants were aged < 15 years, while our study only included those ≥15 years.

A study in India (2009–2011) among hospitalized patients found that including shortness of breath in the WHO case definition for severe acute respiratory illness may underestimate the hospital burden of influenza. Excluding shortness of breath or adding “cough and measured or reported fever” could improve burden estimates [40]. In our study, 12.22% of outpatients in the 2022/23 season and 29.27% in the 2023/24 season had shortness of breath, with sensitivity of 18.07% and 24.14%, and specificity of 88.59% and 70.52%, respectively. Our findings suggest that shortness of breath alone is insufficient to identify all influenza cases, especially mild or atypical ones.

It is important to reiterate that our study was conducted during the COVID-19 pandemic, which may have influenced the findings. Some participants with ILI symptoms tested negative for influenza but positive for COVID-19. A systematic review (Aug 2020–Aug 2021) found fever (63.7%) and cough (51.8%) to be the most prevalent symptoms/signs among COVID-19-positive adults. Most of our patients reported nasal congestion (90% in 2022/23, 85% in 2023/24), a common COVID-19 symptom. However, they also exhibited a range of other symptoms/signs, such as fever, cough, sore throat, headache, and myalgia, many of which overlap with COVID-19. Interestingly, a review noted that headache had a low prevalence (6.7%) among adult COVID-19 patients [41], which is consistent with our findings, where headache was strongly associated with influenza.

Numerous studies have evaluated the usefulness of sore throat as a predictor of influenza, with a consensus supporting its exclusion from clinical case definitions to improve diagnostic performance [7,17,42]. Our findings align with this consensus. There is also ongoing debate about the role of fever as a predictor of influenza. In our study, both fever and cough demonstrated strong diagnostic performance in detecting influenza, a finding corroborated by numerous other studies [7,11,19,34,35,38,40,42].

In order to detect differences by certain characteristics, we compared various variables across types/subtypes of influenza. Regarding symptoms/signs associated with influenza during the 2022/23 season, participants with influenza A(H1N1)pdm09 were slightly more likely to experience a sudden onset of symptoms and myalgia compared to those with influenza A(H3N2). Conversely, participants with A(H1N1)pdm09 were less likely to report headaches than those with A(H3N2), and they were less likely to report malaise compared to participants with influenza B. Patients with influenza A(H3N2) were more likely to experience headaches and dizziness compared to those with influenza B. In contrast, participants with influenza B were more likely to report sore throat, malaise, and chills compared to those with influenza A(H3N2). In the following season (2023/24), only patients presenting with dizziness were significantly more likely to test positive for influenza A(H1N1)pdm09 compared to those with influenza A(H3N2), while other symptoms/signs were similarly distributed by type/subtype of influenza.

The literature provides limited data comparing symptoms/signs across different influenza types and subtypes, particularly among outpatients. A study in Pakistan (2008–2011) with 1489 cases reported that children aged ≤ 5 years and adults aged 21–40 years with A(H1N1) and A(H3N2) more frequently had fever and cough than those with A(H1N1)pdm09 or B. Sore throat was more common in A(H1N1)pdm09 and B infections in children aged ≤ 5 years, and diarrhea was a significant feature in A(H1N1)pdm09 in children under 5 [43]. These differences were mostly observed in younger populations, particularly regarding gastrointestinal symptoms/signs [44]. Another study in Canada (1993–2008) with 1849 cases (both inpatients and outpatients) found that vaccinated patients were less likely to test positive for influenza A than B, and influenza A was more associated with comorbidities like asthma, cardiovascular diseases, COPD, and diabetes, as well as more hospitalizations and physician visits [45]. In our study, which focused on adult outpatients with a median age of 53 years in the 2022/23 season and 54 years in the 2023/24 season, chronic diseases were evenly distributed across all influenza types and subtypes, likely reflecting the study population’s characteristics. On the other hand, some studies have reported no significant differences in the performance of case definitions or symptoms/signs based on the influenza virus type [7,46].

We found that participants who tested positive between December and February were more likely to have influenza A(H1N1)pdm09 or A(H3N2), whereas those who tested positive between February and April were more likely to have influenza B compared to the other two influenza types. This pattern of alternating predominance between influenza A and B types has been described in other studies [47,48,49,50,51].

No single symptom can accurately diagnose influenza, so laboratory confirmation in clinically compatible cases is essential [7]. The goal of influenza surveillance is to monitor transmission, disease burden, and circulating viruses, not to capture every case. A case definition should integrate signs, symptoms, and laboratory results that uniquely characterize influenza, and it should be easy to apply. Sensitivity is crucial for outbreak assessment to minimize missed cases, while high specificity is more important in research settings to minimize false positives [17,34]. The ECDC ILI case definition, which does not require fever, is considered to be balanced for routine surveillance. However, for epidemic scenarios or identifying new strains, more sensitive definitions are necessary. Influenza case definitions should be tailored to the context, prioritizing sensitivity during outbreaks and specificity in routine surveillance [52].

This study had several limitations. First, since our research was conducted during the concurrent circulation of both SARS-CoV-2 and the influenza virus, it is possible that our results were influenced by this co-circulation. Additionally, the low prevalence of laboratory-confirmed influenza cases, particularly in the 2023/24 season (3.8%), affected the performance (sensitivity and PPV) of observed influenza symptoms/signs and their combinations. Second, we did not conduct microbiological tests on influenza-negative cases to detect other respiratory-tract pathogens. Third, due to inconsistencies among participants, some variables, such as smoking and alcohol consumption, may have biased the study results. During the 2022/23 and 2023/24 seasons, smoking was reported by less than 20% and 10% of participants, respectively, while alcohol consumption was reported by less than 4% and 3% in the observed seasons, respectively. Fourth, although we organized training for healthcare providers in proper sampling techniques and sample-handling procedures before the study, it is possible that some swabs were inadequately collected, which could have resulted in false negative results. Fifth, due to the low prevalence of positive cases and the similar distribution of influenza virus types and subtypes across the two seasons, we did not perform validation of the symptoms/signs related to influenza according to virus type. No differences in the performance of the case definitions or the symptoms/signs in influenza cases according to virus type were found in previous studies [7,15]. However, we observed certain differences in the frequency of variables related to the type of influenza virus. Although the potential limitations discussed above are acknowledged, we believe that they did not markedly affect the main results of our study.

## 5. Conclusions

The findings of our study indicated that multiple symptoms/signs related to influenza were good predictors of laboratory-confirmed influenza. The ECDC ILI case definition demonstrated superior discriminatory ability compared to the WHO ILI case definition in this study. With an increasing number of symptoms and signs, the sensitivity of case definitions for influenza decreased, while their specificity increased. Gastrointestinal symptoms/signs most significantly contributed to increased specificity.

In light of our findings, accurate identification of influenza activity still requires laboratory confirmation for some cases. While our results support much of the existing literature, they emphasize the importance of integrating multiple clinical indicators and considering the epidemiological context to enhance diagnostic accuracy. Clinicians and public health officials must consider the specific context when applying case definitions, as decisions regarding testing and isolation depend heavily on these definitions.

Achieving an optimal balance between sensitivity and specificity is crucial for informed and effective clinical and public health decisions. Future research should focus on integrating clinical findings with rapid diagnostic tests and predictive models. This approach will refine diagnostic strategies and ensure timely, accurate influenza identification across different clinical settings.

## Figures and Tables

**Table 1 viruses-17-00272-t001:** Characteristics of participants included in the study during the 2022/23 influenza season.

Characteristics		All Participants (n = 679)		Participants with Laboratory-Confirmed Influenza (n = 83)		Participants Who Tested Negative for Influenza (n = 596)		OR (95% CI)	*p*-Value
		n	(%)	n	(%)	n	(%)		
**Gender**	Male	237	34.90	33	39.76	204	34.23	ref.	
	Female	442	65.10	50	60.24	392	65.77	0.79 (0.49–1.26)	0.3228
**Age (years)**	15–29	112	16.49	28	33.73	84	14.09	12.29 (5.18–29.15)	**<0.0001**
	30–64	302	44.48	48	57.83	254	42.62	6.97 (3.09–15.68)	**<0.0001**
	≥65	265	39.03	7	8.43	258	43.29	ref.	
	Mean age (±standard deviation)	52.84 (±21.27)	NA	37.90 (±17.15)	NA	54.92 (±20.97)	NA		**<0.0001 #**
	Median age (Q1–Q3 interquartile range)	53.0 (36.0–72.0)	NA	37.0 (19.0–49.0)	NA	59.0 (38.0–73.0)	NA		
**Employment status**	Yes *	273	40.21	45	54.22	228	38.26	7.27 (3.22–16.45)	**<0.0001**
	No *	141	20.77	31	37.35	110	18.46	10.39 (4.44–24.30)	**<0.0001**
	Retired **	265	39.03	7	8.43	258	43.29	ref.	
**Marital status**	Unmarried	198	29.16	45	54.22	153	25.67	6.18 (1.85–20.61)	**0.0031**
	Married	395	58.17	32	38.55	363	60.91	1.85 (0.55–6.23)	0.3198
	Widow	66	9.72	3	3.61	63	10.57	ref.	
	Divorced	20	2.95	3	3.61	17	2.85	3.71 (0.69–20.04)	0.1282
**Symptoms and signs *****	Fever (≥38 °C) ****	209	30.78	73	87.95	136	22.82	24.69 (12.41–49.13)	**<0.0001**
	Cough	384	56.55	71	85.54	313	52.52	5.35 (2.84–10.10)	**< 0.0001**
	Sudden onset of symptoms	251	36.97	65	78.31	186	31.21	7.96 (4.59–13.80)	**<0.0001**
	Headache	224	32.99	51	61.45	173	29.03	3.90 (2.42–6.27)	**<0.0001**
	Dizziness	104	15.32	23	27.71	81	13.59	2.44 (1.43–4.16)	**0.0011**
	Sore throat	602	88.66	65	78.31	537	90.10	0.40 (0.22–0.71)	**0.0020**
	Nasal congestion	611	89.99	72	86.75	539	90.44	0.69 (0.35–1.38)	0.2965
	Myalgia	198	29.16	60	72.29	138	23.15	8.66 (5.16–14.51)	**<0.0001**
	Malaise	245	36.08	70	84.34	175	29.36	12.95 (6.99–24.02)	**<0.0001**
	Chills	169	24.89	54	65.06	115	19.30	7.79 (4.75–12.78)	**<0.0001**
	Loss of appetite	81	11.93	20	24.10	61	10.23	2.78 (1.58–4.91)	**0.0004**
	Abdominal pain	51	7.51	8	9.64	43	7.21	1.37 (0.62–3.03)	0.4342
	Nausea	43	6.33	5	6.02	38	6.38	0.94 (0.36–2.46)	0.9019
	Vomiting	24	3.53	7	8.43	17	2.85	3.14 (1.26–7.81)	**0.0140**
	Diarrhea	29	4.27	4	4.82	25	4.19	1.16 (0.39–3.41)	0.7922
	Shortness of breath	83	12.22	15	18.07	68	11.41	1.71 (0.93–3.16)	0.0856
	Clinical signs of pneumonia (auscultatory)	50	7.36	8	9.64	42	7.05	1.41 (0.64–3.11)	0.3991
**Vaccination status**	Vaccinated against seasonal influenza ever before	112	16.49	6	7.23	106	17.79	0.36 (0.15–0.85)	**0.0195**
	Vaccinated against influenza last year	87	12.81	4	4.82	83	13.93	0.31 (0.11–0.88)	**0.0272**
	Vaccinated against influenza this year	89	13.11	5	6.02	84	14.09	0.39 (0.15–0.99)	**0.0484**
	Vaccinated against COVID-19 in a timely manner	414	60.97	35	42.17	379	63.59	0.42 (0.26–0.67)	**0.0002**
**Chronic disease *****	Hypertension	325	47.86	19	22.89	306	51.34	0.28 (0.16–0.48)	**<0.0001**
	Myocardial infarction	31	4.57	1	1.20	30	5.03	0.23 (0.03–1.71)	0.1511
	Cardiac insufficiency	17	2.50	2	2.41	15	2.52	0.96 (0.21–4.30)	0.9533
	Angina pectoris	44	6.48	2	2.41	42	7.05	0.33 (0.08–1.37)	0.1261
	Arrhythmia	52	7.66	2	2.41	50	8.39	0.27 (0.06–1.13)	0.0729
	Stroke	19	2.80	2	2.41	17	2.85	0.84 (0.19–3.71)	0.8190
	Asthma	57	8.39	4	4.82	53	8.89	0.52 (0.18–1.47)	0.2176
	Diabetes mellitus type 1	4	0.59	1	1.20	3	0.50	2.41 (0.25–23.45)	0.4484
	Diabetes mellitus type 2	76	11.19	4	4.82	72	12.08	0.37 (0.13–1.04)	0.0585
	Obesity	52	7.66	4	4.82	48	8.05	0.58 (0.20–1.65)	0.3049
	Other	301	44.33	24	28.92	277	46.48	0.47 (0.28–0.77)	**0.0030**
	Without chronic diseases	216	31.81	47	56.63	169	28.36	3.30 (2.06–5.27)	**<0.0001**
**The number of children aged 7 to 14 in the patient’s family**	0	558	82.18	58	69.88	500	83.89	0.44 (0.17–1.14)	0.0906
	1	92	13.55	19	22.89	73	12.25	0.99 (0.36–2.80)	0.9965
	≥2	29	4.27	6	7.23	23	3.86	ref.	
**The number of children aged 15 to 18 in the patient’s family**	0	622	91.61	75	90.36	547	91.78	1.65 (0.21–12.84)	0.6347
	1	44	6.48	7	8.43	37	6.21	2.27 (0.25–20.37)	0.4639
	≥2	13	1.91	1	1.20	12	2.01	ref.	
**Contact with someone who had influenza-like symptoms seven days before testing**	Yes	338	49.78	67	80.72	271	45.47	5.02 (2.84–8.87)	**<0.0001**
	No	341	50.22	16	19.28	325	54.53	ref.	
**Smoking**	No	548	80.71	63	75.90	485	81.38	0.42 (0.15–1.17)	0.0974
	≤5 years	21	3.09	5	6.02	16	2.68	ref.	
	6–20 years	45	6.63	8	9.64	37	6.21	0.69 (0.20–2.44)	0.5673
	≥21 years	65	9.57	7	8.43	58	9.73	0.39 (0.11–1.38)	0.1433
**Alcohol consumption**	No	656	96.61	75	90.36	581	97.48	0.47 (0.13–1.74)	0.2591
	Sometimes	14	2.06	3	3.61	11	1.85	ref.	
	≤5 years	6	0.88	4	4.82	2	0.34	7.33 (0.88–61.33)	0.0660
	>5 years	3	0.44	1	1.20	2	0.34	1.83 (0.12–27.80)	0.6621
**Use of buses for transportation purposes**	Yes, often	229	33.73	32	38.55	197	33.05	1.12 (0.63–1.98)	0.7087
	Yes, rarely	269	39.62	28	33.73	241	40.44	0.50 (0.44–1.44)	0.4514
	No	181	26.66	23	27.71	158	26.51	ref.	
**Use of taxis for transportation purposes**	Yes, often	64	9.43	5	6.02	59	9.90	ref.	
	Yes, rarely	466	68.63	58	69.88	408	68.46	1.68 (0.65–4.35)	0.2876
	No	149	21.94	20	24.10	129	21.64	1.83 (0.65–5.11)	0.2491
**Previous hospitalizations due to confirmed influenza**	Yes	2	0.29	1	1.20	1	0.17	ref.	
	No	677	99.71	82	98.80	595	99.83	0.14 (0.01–2.22)	0.1626
**Period of inclusion in the study**	1.12.22–14.2.2023.	329	48.45	49	59.04	280	46.98	1.63 (1.02–2.59)	**0.0408**
	15.2.2023–30.4.2023.	350	51.55	34	40.96	316	53.02	ref.	

Note: * for age 15–64 years; ** for age ≥ 65 years; *** one patient could have one or more symptoms/signs or comorbidities simultaneously; **** including feverishness; NA, not applicable; # Student’s *t*-test. Values that differ significantly (*p* < 0.05) are marked in bold.

**Table 2 viruses-17-00272-t002:** Characteristics of participants included in the study during the 2023/24 influenza season.

Characteristics		All Participants (n = 755)		Participants with Laboratory-Confirmed Influenza (n = 29)		Participants Who Tested Negative for Influenza (n = 726)		OR (95% CI)	*p*-Value
		n	(%)	n	(%)	n	(%)		
**Gender**	Male	280	37.09	12	41.38	268	36.91	ref.	
	Female	475	62.91	17	58.62	458	63.09	0.83 (0.39–1.76)	0.6259
**Age (years)**	15–29	105	13.91	13	44.83	92	12.67	9.96 (3.17–31.30)	**<0.0001**
	30–64	364	48.21	12	41.38	352	48.48	2.40 (0.77–7.53)	0.1325
	≥65	286	37.88	4	13.79	282	38.84	ref.	
	Mean age (±standard deviation)	53.63 (±18.73)	NA	40.21 (±20.48)	NA	54.17 (±18.47)	NA		**<0.0001 #**
	Median age (Q1–Q3 interquartile range)	54.0 (40.0–70.0)	NA	38.0 (22.5–59.0)	NA	55.0 (41.0–70.0)	NA		
**Employment status**	Yes *	295	39.07	15	51.72	280	38.57	3.78 (1.24–11.52)	**0.0195**
	No *	174	23.05	10	34.48	164	22.59	4.30 (1.33–13.93)	**0.0150**
	Retired **	286	37.88	4	13.79	282	38.84	ref.	
**Marital status**	Unmarried	188	24.90	17	58.62	171	23.55	1.09 (0.13–8.99)	0.9337
	Married	490	64.90	9	31.03	481	66.25	0.21 (0.02–1.77)	0.1497
	Widow	65	8.61	2	6.90	63	8.68	0.35 (0.03–4.19)	0.4065
	Divorced	12	1.59	1	3.45	11	1.52	ref.	
**Symptoms and signs *****	Fever (≥38 °C) ****	301	39.87	27	93.10	274	37.74	22.27 (5.25–94.39)	**<0.0001**
	Cough	654	86.62	29	100.00	625	86.09	9.57 (0.58–157.93)	0.1142
	Sudden onset of symptoms	268	35.50	20	68.97	248	34.16	4.28 (1.92–9.55)	**0.0004**
	Headache	240	31.79	22	75.86	218	30.03	7.32 (3.08–17.40)	**<0.0001**
	Dizziness	73	9.67	2	6.90	71	9.78	0.68 (0.16–2.93)	0.6085
	Sore throat	437	57.88	20	68.97	417	57.44	1.65 (0.74–3.67)	0.2219
	Nasal congestion	640	84.77	16	55.17	624	85.95	0.20 (0.09–0.43)	**<0.0001**
	Myalgia	327	43.31	15	51.72	312	42.98	1.42 (0.68–2.99)	0.3533
	Malaise	377	49.93	23	79.31	354	48.76	4.03 (1.62–10.01)	**0.0027**
	Chills	141	18.68	6	20.69	135	18.60	1.14 (0.46–2.86)	0.7767
	Loss of appetite	90	11.92	4	13.79	86	11.85	1.19 (0.40–3.50)	0.7513
	Abdominal pain	36	4.77	2	6.90	34	4.68	1.51 (0.34–6.60)	0.5859
	Nausea	48	6.36	3	10.34	45	6.20	1.75 (0.51–5.99)	0.3754
	Vomiting	14	1.85	1	3.45	13	1.79	1.96 (0.25–15.50)	0.5241
	Diarrhea	29	3.84	1	3.45	28	3.86	0.89 (0.12–6.78)	0.9107
	Shortness of breath	221	29.27	7	24.14	214	29.48	0.76 (0.32–1.81)	0.5367
	Clinical signs of pneumonia (auscultatory)	157	20.79	3	10.34	154	21.21	0.43 (0.13–1.43)	0.1693
**Vaccination status**	Vaccinated against seasonal influenza ever before	78	10.33	3	10.34	75	10.33	1.00 (0.30–3.39)	0.9980
	Vaccinated against influenza last year	68	9.01	3	10.34	65	8.95	1.17 (0.35–3.98)	0.7976
	Vaccinated against influenza this year	71	9.40	3	10.34	68	9.37	1.12 (0.33–3.79)	0.8596
	Vaccinated against COVID-19 in a timely manner	469	62.12	13	44.83	456	62.81	0.48 (0.23–1.02)	0.0549
**Chronic disease *****	Hypertension	343	45.43	7	24.14	336	46.28	0.37 (0.16–0.88)	**0.0237**
	Myocardial infarction	22	2.91	1	3.45	21	2.89	1.20 (0.16–9.23)	0.8617
	Cardiac insufficiency	26	3.44	1	3.45	25	3.44	1.00 (0.13–7.66)	0.9989
	Angina pectoris	16	2.12	1	3.45	15	2.07	1.69 (0.22–13.27)	0.6163
	Arrhythmia	32	4.24	1	3.45	31	4.27	0.80 (0.11–6.08)	0.8298
	Stroke	12	1.59	0	0.00	12	1.65	0.97 (0.06–16.76)	0.9826
	Asthma	63	8.34	3	10.34	60	8.26	1.28 (0.38–4.36)	0.6919
	Diabetes mellitus type 1	11	1.46	1	3.45	10	1.38	2.56 (0.32–20.68)	0.3786
	Diabetes mellitus type 2	66	8.74	0	0.00	66	9.09	2.56 (0.32–20.68)	0.3786
	Obesity	50	6.62	3	10.34	47	6.47	1.67 (0.49–5.71)	0.4159
	Other	206	27.28	6	20.69	200	27.55	0.69 (0.28–1.71)	0.4187
	Without chronic diseases	318	42.12	17	58.62	301	41.46	2.00 (0.94–4.25)	0.0714
**Previous hospitalizations due to confirmed influenza**	0	698	92.45	27	93.10	671	92.42	1.19 (0.07–20.43)	0.9057
	1	43	5.70	2	6.90	41	5.65	1.75 (0.08–38.58)	0.7238
	≥2	14	1.85	0	0.00	14	1.93	ref.	
**The number of children aged 15 to 18 in the patient’s family**	0	709	93.91	28	96.55	681	93.80	1.55 (0.09–26.32)	0.7627
	1	28	3.71	1	3.45	27	3.72	2.02 (0.08–52.28)	0.6724
	≥2	18	2.38	0	0.00	18	2.48	ref.	
**Contact with someone who had influenza-like symptoms seven days before testing**	Yes	602	79.74	25	86.21	577	79.48	1.61 (0.55–4.71)	0.3809
	No	153	20.26	4	13.79	149	20.52	ref.	
**Smoking**	No	684	90.60	25	86.21	659	90.77	2.05 (0.12–4.58)	0.6187
	≤5 years	7	0.93	2	6.90	5	0.69	24.09 (1.01–574.85)	**0.0493**
	6–20 years	26	3.44	0	0.00	26	3.58	ref.	
	≥21 years	38	5.03	2	6.90	36	4.96	3.63 (0.17–78.78)	0.4116
**Alcohol consumption**	No	737	97.62	29	100.00	708	97.52	NA	
	Sometimes	3	0.40	0	0.00	3	0.41		
	≤5 years	1	0.13	0	0.00	1	0.14		
	>5 years	14	1.85	0	0.00	14	1.93		
**Use of buses for transportation purposes**	Yes, often	326	43.18	14	48.28	312	42.98	0.83 (0.29–2.38)	0.7351
	Yes, rarely	331	43.84	10	34.48	321	44.21	0.58 (0.19–1.74)	0.3300
	No	98	12.98	5	17.24	93	12.81	ref.	
**Use of taxis for transportation purposes**	Yes, often	29	3.84	2	6.90	27	3.72	ref.	
	Yes, rarely	611	80.93	19	65.52	592	81.54	0.43 (0.10–1.96)	0.2768
	No	115	15.23	8	27.59	107	14.74	1.01 (0.20–5.03)	0.9909
**Previous hospitalizations due to confirmed influenza**	Yes	0	0.00	0	0.00	0	0.00	NA	
	No	755	100.00	29	100.00	726	100.00		
**Period of inclusion in the study**	1.12.23–14.2.2024.	466	61.72	25	86.21	441	60.74	4.04 (1.39–11.73)	**0.0103**
	15.2.2024–30.4.2024.	289	38.28	4	13.79	285	39.26	ref.	

Note: * for age 15–64 years; ** for age ≥ 65 years; *** one patient could have one or more symptoms/signs or comorbidities simultaneously; **** including feverishness; NA, not applicable; # Student’s *t*-test. Values that differ significantly (*p* < 0.05) are marked in bold.

**Table 3 viruses-17-00272-t003:** Performance of symptoms/signs related to influenza in the 2022/23 and 2023/24 influenza seasons.

Season	Symptoms and Signs	Se % (95% CI)	Sp % (95% CI)	PPV % (95% CI)	NPV % (95% CI)	LR+ (95% CI)	LR− (95% CI)	Accuracy (95% CI)	Kappa (95% CI)
**2022/23**	Fever (≥38 °C) *	87.95 (78.96–94.07)	77.18 (73.60–80.49)	34.93 (31.22–38.83)	97.87 (96.25–98.80)	3.85 (3.26–4.56)	0.16 (0.09–0.28)	0.785 (0.763–0.799)	0.394 (0.333–0.433)
	Cough	85.54 (76.11–92.30)	47.48 (43.41–51.58)	18.49 (16.79–20.32)	95.93 (93.28–97.57)	1.63 (1.45–1.83)	0.30 (0.18–0.52)	0.521 (0.499–0.537)	0.129 (0.087–0.157)
	Sudden onset of symptoms	78.31 (67.91–86.61)	68.79 (64.90–72.49)	25.90 (22.87–29.17)	95.79 (93.78–97.18)	2.51 (2.13–2.96)	0.32 (0.21–0.48)	0.700 (0.675–0.719)	0.252 (0.191–0.299)
	Headache	61.45 (50.12–71.93)	70.97 (67.15–74.59)	22.77 (19.26–26.70)	92.97 (90.93–94.57)	2.12 (1.71–2.62)	0.54 (0.41–0.72)	0.698 (0.672–0.722)	0.187 (0.118–0.251)
	Dizziness	27.71 (18.45–38.62)	86.41 (83.39–89.06)	22.12 (15.96–29.80)	89.57 (88.21–90.78)	2.04 (1.36–3.05)	0.84 (0.73–0.96)	0.792 (0.772–0.816)	0.127 (0.041–0.227)
	Sore throat	78.31 (67.91–86.61)	9.90 (7.62–12.58)	10.80 (9.73–11.97)	76.62 (67.08–84.06)	0.87 (0.77–0.98)	2.19 (1.36–3.52)	0.183 (0.160–0.201)	−0.032 (−0.060; −0.009)
	Nasal congestion	86.75 (77.52–93.19)	9.56 (7.32–12.21)	11.78 (10.90–12.73)	83.82 (73.92–90.45)	0.96 (0.88–1.05)	1.39 (0.76–2.53)	0.190 (0.170–0.205)	−0.010 (−0.035–0.008)
	Myalgia	72.29 (61.38–81.55)	76.85 (73.25–80.18)	30.30 (26.29–34.64)	95.22 (93.35–96.58)	3.12 (2.56–3.80)	0.36 (0.25–0.51)	0.763 (0.738–0.784)	0.308 (0.235–0.369)
	Malaise	84.34 (74.71–91.39)	70.64 (66.80–74.27)	28.57 (25.51–31.84)	97.00 (95.15–98.16)	2.87 (2.46–3.35)	0.22 (0.13–0.37)	0.723 (0.700–0.739)	0.299 (0.241–0.340)
	Chills	65.06 (53.81–75.20)	80.70 (77.30–83.80)	31.95 (27.22–37.09)	94.31 (92.50–95.71)	3.37 (2.69–4.23)	0.43 (0.32–0.58)	0.788 (0.763–0.811)	0.317 (0.235–0.389)
	Loss of appetite	24.10 (15.38–34.73)	89.77 (87.05–92.08)	24.69 (17.29–33.95)	89.46 (88.24–90.58)	2.35 (1.50–3.69)	0.85 (0.75–0.96)	0.817 (0.798–0.840)	0.140 (0.049–0.246)
	Abdominal pain	9.64 (4.25–18.11)	92.79 (90.40–94.73)	15.69 (8.31–27.63)	88.06 (87.26–88.81)	1.34 (0.65–2.74)	0.97 (0.90–1.05)	0.826 (0.814–0.845)	0.029 (−0.039–0.133)
	Nausea	6.02 (1.98–13.50)	93.62 (91.35–95.45)	11.63 (5.06–24.52)	87.74 (87.09–88.35)	0.94 (0.38–2.33)	1.00 (0.95–1.06)	0.829 (0.820–0.846)	−0.004 (−0.058–0.095)
	Vomiting	8.43 (3.46–16.61)	97.15 (95.47–98.33)	29.17 (14.97–49.06)	88.40 (87.69–89.06)	2.96 (1.26–6.92)	0.94 (0.88–1.01)	0.863 (0.852–0.878)	0.080 (0.007–0.181)
	Diarrhea	4.82 (1.33–11.88)	95.81 (93.87–97.27)	13.79 (5.40–30.95)	87.85 (87.29–88.38)	1.15 (0.41–3.22)	0.99 (0.94–1.05)	0.847 (0.839–0.862)	0.009 (−0.042–0.108)
	Shortness of breath	18.07 (10.48–28.05)	88.59 (85.76–91.03)	18.07 (1.70–26.86)	88.59 (87.48–89.61)	1.58 (0.95–2.64)	0.92 (0.83–1.03)	0.800 (0.783–0.822)	0.067 (−0.013–0.169)
	Clinical signs of pneumonia (auscultatory)	9.64 (4.25–18.11)	92.95 (90.59–94.87)	16.00 (8.48–28.13)	88.08 (87.28–88.83)	1.37 (0.67–2.81)	0.97 (0.90–1.05)	0.828 (0.816–0.846)	0.031 (−0.037–0.136)
**2023/24**	Fever (≥38 °C) *	93.10 (77.23–99.15)	62.26 (58.62–65.80)	8.97 (7.92–10.15)	99.56 (98.34–99.88)	2.47 (2.15–2.83)	0.11 (0.03–0.42)	0.634 (0.621–0.639)	0.101 (0.068–0.111)
	Cough	100.00 (88.06–100.00)	13.91 (11.48–16.64)	4.43 (4.31–4.56)	100.00 (96.41–100.00)	1.16 (1.13–1.20)	0	0.172 (0.161–0.172)	0.012 (−0.001–0.012)
	Sudden onset of symptoms	68.97 (49.17–84.72)	65.84 (62.26–69.29)	7.46 (5.83–9.50)	98.15 (96.85–98.92)	2.02 (1.55–2.63)	0.47 (0.27–0.81)	0.660 (0.645–0.671)	0.070 (0.029–0.102)
	Headache	75.86 (56.46–89.70)	69.97 (66.49–73.29)	9.17 (7.40–11.30)	98.64 (97.44–99.28)	2.53 (2.00–3.19)	0.34 (0.18–0.66)	0.702 (0.687–0.712)	0.102 (0.057–0.132)
	Dizziness	6.90 (0.85–22.77)	90.22 (87.82–92.28)	2.74 (0.72–9.85)	96.04 (95.63–96.41)	0.71 (0.18–2.74)	1.03 (0.93–1.14)	0.870 (0.866–0.883)	−0.017 (−0.051–0.083)
	Sore throat	68.97 (49.17–84.72)	42.56 (38.93–46.25)	4.58 (3.59–5.81)	97.17 (95.20–98.35)	1.20 (0.93–1.54)	0.73 (0.42–1.26)	0.436 (0.421–0.447)	0.015 (−0.011–0.035)
	Nasal congestion	55.17 (35.69–73.55)	14.05 (11.60–16.79)	2.50 (1.81–3.44)	88.70 (83.45–92.43)	0.64 (0.46–0.89)	3.19 (2.05–4.96)	0.156 (0.142–0.170)	−0.028 (−0.045; −0.011)
	Myalgia	51.72 (32.53–70.55)	57.02 (53.33–60.66)	4.59 (3.24–6.46)	96.73 (95.28–97.74)	1.20 (0.84–1.73)	0.85 (0.58–1.24)	0.568 (0.554–0.582)	0.015 (−0.018–0.047)
	Malaise	79.31 (60.28–92.01)	51.24 (47.54–54.93)	6.10 (5.05–7.35)	98.41 (96.80–99.22)	1.63 (1.33–1.99)	0.40 (0.20–0.83)	0.523 (0.508–0.532)	0.045 (0.016–0.064)
	Chills	20.69 (7.99–39.72)	81.40 (78.38–84.17)	4.26 (2.10–8.43)	96.25 (95.51–96.88)	1.11 (0.54–2.31)	0.97 (0.81–1.18)	0.791 (0.782–0.805)	0.007 (−0.036–0.076)
	Loss of appetite	13.79 (3.89–31.66)	88.15 (85.58–90.41)	4.44 (1.80–10.56)	96.24 (95.67–96.74)	1.16 (0.46–2.95)	0.98 (0.84–1.13)	0.853 (0.846–0.867)	0.010 (−0.038–0.103)
	Abdominal pain	6.90 (0.85–22.77)	95.32 (93.52–96.74)	5.56 (1.46–18.90)	96.24 (95.86–96.59)	1.47 (0.37–5.84)	0.98 (0.88–1.08)	0.919 (0.915–0.931)	0.020 (−0.033–0.167)
	Nausea	10.34 (2.19–27.35)	93.80 (91.79–95.44)	6.25 (2.15–16.80)	96.32 (95.85–96.74)	1.67 (0.55–5.06)	0.96 (0.84–1.08)	0.906 (0.900–0.919)	0.032 (−0.029–0.165)
	Vomiting	3.45 (0.09–17.76)	98.21 (96.96–99.04)	7.14 (1.03–36.24)	96.22 (95.96–96.47)	1.93 (0.26–14.23)	0.98 (0.92–1.05)	0.946 (0.943–0.956)	0.022 (−0.023–0.202)
	Diarrhea	3.45 (0.09–17.76)	96.14 (94.47–97.42)	3.45 (0.50–20.22)	96.14 (95.87–96.40)	0.89 (0.13–6.35)	1.00 (0.94–1.08)	0.926 (0.923–0.937)	−0.004 (−0.038–0.147)
	Shortness of breath	24.14 (10.30–43.54)	70.52 (67.06–73.82)	3.17 (1.67–5.92)	95.88 (94.96–96.64)	0.82 (0.43–1.58)	1.08 (0.87–1.33)	0.687 (0.677–0.702)	−0.013 (−0.045–0.035)
	Clinical signs of pneumonia (auscultatory)	10.34 (2.19–27.35)	78.79 (75.63–81.71)	1.91 (0.66–5.43)	95.65 (95.08–96.16)	0.49 (0.17–1.44)	1.14 (1.00–1.29)	0.762 (0.756–0.775)	−0.035 (−0.060–0.024)

Note: Se: sensitivity; Sp: specificity; PPV: positive predictive value; NPV: negative predictive value; LR+: positive likelihood ratio; LR−: negative likelihood ratio; * including feverishness.

**Table 4 viruses-17-00272-t004:** Performance of symptoms/signs related to influenza and case definitions of ILI during the two seasons (2022/23 and 2023/24), by age group.

**Symptoms/Signs 2022/23**	**15–29 (n = 112)**					**30–64 (n = 302)**					**≥65 (n = 265)**					**Total (n = 679)**				
	**Positive (n = 28)**	**Negative (n = 84)**	**Se % (95% CI)**	**Sp % (95% CI)**	**Accuracy (95% CI)**	**Positive (n = 48)**	**Negative (n = 254)**	**Se % (95% CI)**	**Sp % (95% CI)**	**Accuracy (95% CI)**	**Positive (n = 7)**	**Negative (n = 258)**	**Se % (95% CI)**	**Sp % (95% CI)**	**Accuracy (95% CI)**	**Positive** **(n = 83)**	**Negative** **(n = 596)**	**Se % (95% CI)**	**Sp % (95% CI)**	**Accuracy** **(95% CI)**
**WHO ILI**	3	3	10.71 (2.27–28.23)	96.43 (89.92–99.26)	0.750 (0.712–0.788)	5	9	10.42 (3.47–22.66)	96.46 (93.38–98.37)	0.828 (0.808–0.853)	1	11	14.29 (0.36–57.87)	95.74 (92.50–97.85)	0.936 (0.929–0.957)	9	23	10.84 (5.08–19.59)	96.14 (94.27–97.54)	0.857 (0.844–0.874)
**WHO ILI plus sudden onset of symptoms**	22	30	78.57 (59.05–91.70)	64.29 (53.08–74.45)	0.679 (0.591–0.738)	30	44	62.50 (47.35–76.05)	82.68 (77.45–87.12)	0.795 (0.752–0.833)	4	17	57.14 (18.41–90.10)	93.41 (89.66–96.12)	0.925 (0.905–0.941)	56	91	67.47 (56.30–77.35)	84.73 (81.59–87.53)	0.826 (0.801–0.848)
**ECDC ILI**	24	47	85.71 (67.33–95.97)	44.05 (33.22–55.30)	0.545 (0.461–0.592)	35	75	72.92 (58.15–84.72)	70.47 (64.45–76.01)	0.709 (0.665–0.743)	4	29	57.14 (18.41–90.10)	88.76 (84.26–92.34)	0.879 (0.860–0.896)	63	151	75.90 (65.27–84.62)	74.66 (70.97–78.11)	0.748 (0.724–0.768)
**Fever (including feverishness) + headache + malaise + myalgia plus cough**	16	14	57.14 (37.18–75.54)	83.33 (73.62–90.58)	0.768 (0.684–0.842)	13	18	27.08 (15.28–41.85)	92.91 (89.03–95.75)	0.825 (0.792–0.859)	1	6	14.29 (0.36–57.87)	97.67 (95.01–99.14)	0.955 (0.948–0.974)	30	38	36.14 (25.88–47.43)	93.62 (91.35–95.45)	0.866 (0.845–0.887)
**Fever (including feverishness)+ headache + malaise + myalgia plus sore throat**	16	12	57.14 (37.18–75.54)	85.71 (76.38–92.39)	0.786 (0.703–0.857)	11	18	22.92 (12.03–37.31)	92.91 (89.03–95.75)	0.818 (0.787–0.852)	1	4	14.29 (0.36–57.87)	98.45 (96.08–99.58)	0.962 (0.955–0.980)	28	34	33.73 (23.72–44.95)	94.30 (92.12–96.02)	0.869 (0.848–0.890)
**Fever (including feverishness) + headache + malaise + myalgia plus shortness of breath**	3	1	10.71 (2.27–28.23)	98.81 (93.54–99.97)	0.768 (0.730–0.785)	4	4	8.33 (2.32–19.98)	98.43 (96.02–99.57)	0.841 (0.824–0.858)	0	3	0.00 (0.00–40.96)	98.84 (96.64–99.76)	0.962 (0.962–0.977)	7	8	8.43 (3.46–16.61)	98.66 (97.37–99.42)	0.876 (0.866–0.888)
**Symptoms/signs 2023/24**	**15–29 (n = 105)**					**30–64 (n = 364)**					**≥65 (n = 286)**					**Total (n = 755)**				
	**Positive (n = 13)**	**Negative (n = 92)**	**Se % (95% CI)**	**Sp % (95% CI)**	**Accuracy (95% CI)**	**Positive (n = 12)**	**Negative (n = 352)**	**Se % (95% CI)**	**Sp % (95% CI)**	**Accuracy (95% CI)**	**Positive (n = 4)**	**Negative (n = 282)**	**Se % (95% CI)**	**Sp % (95% CI)**	**Accuracy (95% CI)**	**Positive (n = 29)**	**Negative (n = 726)**	**Se % (95% CI)**	**Sp % (95% CI)**	**Accuracy (95% CI)**
**WHO ILI**	6	20	46.15 (19.22–74.87)	78.26 (68.44–86.19)	0.743 (0.681–0.808)	1	53	8.33 (0.21–38.48)	84.94 (80.77–88.51)	0.824 (0.819–0.845)	1	21	25.00 (0.63–80.59)	92.55 (88.84–95.33)	0.916 (0.909–0.931)	8	94	27.59 (12.73–47.24)	87.05 (84.39–89.41)	0.848 (0.837–0.862)
**WHO ILI plus sudden onset of symptoms**	6	25	46.15 (19.22–74.87)	72.83 (62.55–81.58)	0.695 (0.634–0.761)	11	86	91.67 (61.52–99.79)	75.57 (70.73–79.97)	0.761 (0.740–0.766)	2	28	50.00 (6.76–93.24)	90.07 (85.97–93.30)	0.895 (0.884–0.906)	19	139	65.52 (45.67–82.06)	80.85 (77.80–83.65)	0.803 (0.788–0.815)
**ECDC ILI**	7	38	53.85 (25.13–80.78)	58.70 (47.95–68.87)	0.581 (0.515–0.643)	11	127	91.67 (61.52–99.79)	63.92 (58.66–68.94)	0.648 (0.628–0.654)	2	42	50.00 (6.76–93.24)	85.11 (80.41–89.05)	0.846 (0.835–0.858)	20	207	68.97 (49.17–84.72)	71.49 (68.05–74.75)	0.714 (0.699–0.725)
**Fever (including feverishness) + feverishness + headache + malaise + myalgia plus cough**	2	5	15.38 (1.92–45.45)	94.57 (87.77–98.21)	0.848 (0.816–0.899)	2	26	16.67 (2.09–48.41)	92.61 (89.36–95.12)	0.901 (0.892–0.921)	0	5	0.00 (0.00–60.24)	98.23 (95.91–99.42)	0.969 (0.969–0.984)	4	36	13.79 (3.89–31.66)	95.04 (93.20–96.50)	0.919 (0.912–0.932)
**Fever (including feverishness) + headache + malaise + myalgia plus sore throat**	1	3	7.69 (0.19–36.03)	96.74 (90.77–99.32)	0.857 (0.839–0.897)	1	18	8.33 (0.21–38.48)	94.89 (92.04–96.94)	0.920 (0.915–0.940)	0	3	0.00 (0.00–60.24)	98.94 (96.92–99.78)	0.976 (0.976–0.989)	2	24	6.90 (0.85–22.77)	96.69 (95.12–97.87)	0.932 (0.928–0.944)
**Fever (including feverishness) + headache + malaise + myalgia plus shortness of breath**	1	1	7.69 (0.19–36.03)	98.91 (94.09–99.97)	0.876 (0.858–0.894)	1	12	8.33 (0.21–38.48)	96.59 (94.12–98.23)	0.937 (0.932–0.955)	0	1	0.00 (0.00–60.24)	98.94 (96.92–99.78)	0.983 (0.983–0.989)	2	14	6.90 (0.85–22.77)	98.07 (96.79–98.94)	0.946 (0.941–0.956)

Note: Se: sensitivity; Sp: specificity.

## Data Availability

The original contributions presented in this study are included in the article/Appendix A; further inquiries can be directed to the corresponding author.

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
