# Peer review of "Diagnostic Significance of Influenza Symptoms and Signs, and Their Variation by Type/Subtype, in Outpatients Aged ≥ 15 Years: Novi Sad, Serbia"

_viruses, 2025, doi:10.3390/v17020272_

Round 1
Reviewer 1 Report
Comments and Suggestions for Authors
Comments and Suggestions for Authors
The article “Diagnostic Significance of Symptoms and Signs of Influenza in a Population Aged ≥15 Years: Novi Sad, Serbia” presents
the results of an observational prospective study on influenza carried out at primary care level in Novi Sad Serbia, during two influenza seasons (2022/23, 2023/24), in a population aged ≥15 Years.
This study presents the clinical factors associated with laboratory-confirmed influenza and the performance of influenza case definitions in patients aged ≥15 years, using data obtained from a primary healthcare influenza surveillance system.
The topic of research is relevant. The authors use surveillance data and advanced statistical tests; also the references are relevant. Nevertheless, the article requires major revision before accepting for publication.
Comments and questions:
In my opinion, the presented study has 2 very ambitious objectives:
- to identify the signs, symptoms and other factors suggestive of influenza (predictive factors), to be used by the family doctor for the diagnosis of influenza, treatment and specific measures.
- evaluate the performance of different case definitions (WHO vs ECDC):
As a result, there are a lot of epidemiological data and information that make the article difficult to read and understand.
The introduction is very long, with a lot of general information about the influenza. I recommend to be restructured, keeping only the information relevant to the content of the article. The addition of national influenza surveillance data from the two seasons (2022/2023, 2023/2024) would help to better understand the characteristics of influenza in Serbia.
The aim of the study is presented differently in the abstract, introduction, and discussions. It can be improved, redefined to reflect the results presented in the study and the same throughout the article.
Also the title can be improve.
It is not explained why the population over 15 years of age was included in the study and not the entire population.
It is not clear how many family doctors participated in the study (all or those who wanted to?). Were there selection criteria?
I recommend using the term influenza throughout the article (instead of flu).
I recommend using the term “training” (instead of “education”), line 179.
Results
A lot of data, presented both in text and in tables.
Line 223 and 274 - To be filled in with the number of patients. The number of samples is written in the text, but then the characteristics of the patients are presented.
I propose to change “sore throat was significantly higher... “ with “sore throat was significantly associated with a negative result for influenza”, line 244.
Discussion
The discussion part is too long. I recommend keeping only what is relevant to the objectives and the results of this study.
Author Response
Comments and Suggestions for Authors
The article “Diagnostic Significance of Symptoms and Signs of Influenza in a Population Aged ≥15 Years: Novi Sad, Serbia” presents the results of an observational prospective study on influenza carried out at primary care level in Novi Sad Serbia, during two influenza seasons (2022/23, 2023/24), in a population aged ≥15 Years.
This study presents the clinical factors associated with laboratory-confirmed influenza and the performance of influenza case definitions in patients aged ≥15 years, using data obtained from a primary healthcare influenza surveillance system.
The topic of research is relevant. The authors use surveillance data and advanced statistical tests; also the references are relevant. Nevertheless, the article requires major revision before accepting for publication.
Authors’ response: Thank you very much for the careful reading of the manuscript and your constructive suggestions. We have revised our manuscript according to your comments and suggestions point by point (please see the revised manuscript and following answers).
Comments and questions:
In my opinion, the presented study has 2 very ambitious objectives:
- to identify the signs, symptoms and other factors suggestive of influenza (predictive factors), to be used by the family doctor for the diagnosis of influenza, treatment and specific measures.
- evaluate the performance of different case definitions (WHO vs ECDC):
As a result, there are a lot of epidemiological data and information that make the article difficult to read and understand.
The introduction is very long, with a lot of general information about the influenza. I recommend to be restructured, keeping only the information relevant to the content of the article. The addition of national influenza surveillance data from the two seasons (2022/2023, 2023/2024) would help to better understand the characteristics of influenza in Serbia.
Authors’ response: Thank you for these comments. We have revised the Introduction section; however, due to the absence of national-level influenza surveillance data for the 2022/23 and 2023/24 seasons, we did not include this information in the Introduction. In this context, the aim of our study was not to analyze influenza-like illness or acute respiratory infection surveillance but rather to predict influenza among participants aged ≥15 years based on their characteristics, as presented in the Results section.
The aim of the study is presented differently in the abstract, introduction, and discussions. It can be improved, redefined to reflect the results presented in the study and the same throughout the article.
Authors’ response: Thank you for these comments. We have revised the sentences outlining the study aim in the Abstract, Introduction, and Discussion sections according to your suggestions.
Also the title can be improve.
Authors’ response: Thank you for this suggestion. However, we would like to emphasize that this study is part of an unpublished PhD thesis with the same title. To ensure that the article accurately reflects the process evaluation of the PhD thesis, we believe it is preferable to retain the current title of this paper.
It is not explained why the population over 15 years of age was included in the study and not the entire population.
Authors’ response: Thank you for your consideration. As stated in the Introduction, previous research has evaluated the diagnostic validity of clinical features associated with influenza, with a stronger focus on pediatric populations than on adults. In adult populations, there is a paucity of data on how pre-existing health status, vaccination history, or comorbid conditions affect the diagnostic accuracy of case definitions.
Furthermore, as mentioned in the Introduction, only one prior study in our country has assessed the diagnostic accuracy of the WHO ILI case definition. This study included both ambulatory and hospitalized patients of all ages but did not consider pre-existing conditions, vaccination status, or contact history. Its findings demonstrated limited diagnostic utility, particularly in older adults, with low specificity and a high rate of false positives. Additionally, the study did not examine whether specific symptoms were more predictive of influenza type A or B or whether they varied by viral subtype [Ref. No 22: Ristić M, Petrović V. Evaluation of the diagnostic utility of case definitions to detect influenza virus infection in Vojvodina, Serbia. Srp Arh Celok Lek. 2020;148(1-2):100-05.].
Considering all of the above, our surveillance at the primary care level was conducted exclusively among general practitioners (GPs) rather than pediatricians. Consequently, we included only patients aged ≥15 years. Additionally, in the Materials and Methods section (Study Design), we have clarified that participant enrollment was conducted by general practitioners at the primary care level.
It is not clear how many family doctors participated in the study (all or those who wanted to?). Were there selection criteria?
Authors’ response: Thank you for this insightful question. The surveillance was designed to include five general practitioners (GPs) without specific selection criteria, except that these GPs had previously participated in influenza-like illness surveillance during past influenza seasons. However, only during the 2022/23 and 2023/24 seasons did they also collect patient specimens for laboratory testing for influenza. All of them received training on the inclusion and exclusion criteria for participant enrollment in this study.
I recommend using the term influenza throughout the article (instead of flu).
Authors’ response: Thank you for this recommendation. We have replaced all instances of “flu” with “influenza” throughout the paper.
I recommend using the term “training” (instead of “education”), line 179.
Authors’ response: Thank you for this recommendation. We have replaced the term “education” with “training”.
Results
A lot of data, presented both in text and in tables.
Authors’ response: We fully agree with your comments. However, as mentioned above, this paper represents only a part of an unpublished PhD thesis, which includes more/additional results. Although our aim was to present only the most significant findings, a large amount of data remains in the paper. To maintain focus on the specific results, we have organized the paper so that other results are presented in the Supplementary section for readers who may be more interested. We believe that presenting our results in this manner will enable future researchers to review, compare, and evaluate our findings for their own studies.
Line 223 and 274 - To be filled in with the number of patients. The number of samples is written in the text, but then the characteristics of the patients are presented.
Authors’ response: Thank you for these comments. We have corrected as “participant samples” instead of “samples”.
I propose to change “sore throat was significantly higher... “ with “sore throat was significantly associated with a negative result for influenza”, line 244.
Authors’ response: Thank you for this proposal. We have corrected the “Sore throat was significantly associated with a negative result for influenza”' instead of “Sore throat was significantly higher among influenza negative compared to influenza positive cases”.
Discussion
The discussion part is too long. I recommend keeping only what is relevant to the objectives and the results of this study.
Authors’ response: We fully agree with your comments once again. However, with your consent to retain the results of this study, we believe it is reasonable for the majority of the discussion to remain in its current form. As outlined in the Discussion section, we have focused solely on the statistically significant results of our study. Additionally, we have made an effort to explain why our findings differ from those of other relevant studies, while also highlighting similarities with the results of other researchers in this field. While we value the importance of your suggestion, we have streamlined the discussion to preserve the integrity of the key findings, avoiding unnecessary complexity in the interpretation. We believe that further shortening of the discussion could jeopardize the presentation of the main findings of our work.
Reviewer 2 Report
Comments and Suggestions for Authors
The manuscript of “Diagnostic Significance of Symptoms and Signs of Influenza in a Population Aged ≥15 Years: Novi Sad, Serbia” assessed the diagnostic performance of symptoms and signs related to influenza virus infection in adults. The study showed that the younger adults are more susceptible to influenza virus than the older, and the fever, cough, and sudden onset of symptoms are most strongly associated with influenza infection,which could be the most reliable predictor of influenza infection.
1. In this study,679 patients with flu-like symptom were analyzed,while only 83 (12.2%) tested positive for influenza virus infection, the positive rate is low, which suggested that some other pathogens such as COVID19 might co-circulate with the flu during the investigation period, thus the influence of the other disease should be fully analyzed and discussed in this study, including the positive rate of the COVID19 in the 679 patients .
2.This study concluded that the fever is the most reliable predictor of influenza infection, while based on Table2, 301 patients have fever, but only 27 out of them are flu positive, which indicated that the most of the patients with fever were not infected by influenza virus. Althoug most of the influenza virus associated patients have fever, it might not be a perfect predictors of influenza infection.
Author Response
The manuscript of “Diagnostic Significance of Symptoms and Signs of Influenza in a Population Aged ≥15 Years: Novi Sad, Serbia” assessed the diagnostic performance of symptoms and signs related to influenza virus infection in adults. The study showed that the younger adults are more susceptible to influenza virus than the older, and the fever, cough, and sudden onset of symptoms are most strongly associated with influenza infection, which could be the most reliable predictor of influenza infection.
Authors’ response: Thank you for your appreciation and for your constructive comments, which have been addressed accordingly. We have thoroughly revised our manuscript based on your comments and suggestions. Below is a point-by-point response detailing the revisions made:
1. In this study,679 patients with flu-like symptom were analyzed, while only 83 (12.2%) tested positive for influenza virus infection, the positive rate is low, which suggested that some other pathogens such as COVID19 might co-circulate with the flu during the investigation period, thus the influence of the other disease should be fully analyzed and discussed in this study, including the positive rate of the COVID19 in the 679 patients .
Authors’ response: Thank you for this suggestion. As stated in the objectives of our study, we evaluated the diagnostic performance of symptoms and signs associated with influenza, both individually and in combination, as well as differences in patient characteristics based on the type and subtype of influenza. Our study was conducted during the 2022/23 and 2023/24 influenza seasons, a period during which COVID-19 was also circulating in the population. However, due to limitations in diagnostic testing, we only performed laboratory testing on specimens from participants with symptoms or signs suggestive of influenza-like illness for influenza viruses, and not for other respiratory pathogens. We acknowledge that the low prevalence of influenza among tested participants may be explained by the co-circulation of other respiratory pathogens. This issue was addressed in the Discussion section, and we also recognized it as one of the limitations of our study. Considering this limitation, future research may focus on refining the case definitions for influenza and COVID-19.
2. This study concluded that the fever is the most reliable predictor of influenza infection, while based on Table2, 301 patients have fever, but only 27 out of them are flu positive, which indicated that the most of the patients with fever were not infected by influenza virus. Althoug most of the influenza virus associated patients have fever, it might not be a perfect predictors of influenza infection.
Authors’ response: Thank you for this suggestion. The results of our research showed that fever (≥ 38°C) was present in 30.78% of participants during the 2022/23 influenza season and in 39.87% during the 2023/24 season. Fever was not a mandatory symptom for inclusion in the study. Accordingly, we found that participants with fever were more likely to have influenza compared to those without fever. During the 2022/23 season, 87.95% of participants with laboratory-confirmed influenza had fever, while 22.82% of influenza-negative participants also had fever (p<0.0001). Similarly, in the 2023/24 season, 93.10% of participants with laboratory-confirmed influenza had fever, compared to 37.74% of influenza-negative participants (p<0.0001) (Tables 1 and 2).
Round 2
Reviewer 1 Report
Comments and Suggestions for Authors
Comments and Suggestions for Authors
The article “Diagnostic Significance of Symptoms and Signs of Influenza in a Population Aged ≥15 Years: Novi Sad, Serbia” presents the results of an observational prospective study on influenza carried out at primary care level in Novi Sad Serbia, during two influenza seasons (2022/23, 2023/24), in a population aged ≥15 Years.
This study presents the clinical factors associated with laboratory-confirmed influenza and the performance of influenza case definitions in patients aged ≥15 years, using data obtained from a primary healthcare influenza surveillance system.
The topic of research is relevant. The authors use surveillance data and advanced statistical tests; also the references are relevant. Nevertheless, the article requires major revision before accepting for publication.
Authors’ response: Thank you very much for the careful reading of the manuscript and your constructive suggestions. We have revised our manuscript according to your comments and suggestions point by point (please see the revised manuscript and following answers).
Comments and questions:
In my opinion, the presented study has 2 very ambitious objectives:
- to identify the signs, symptoms and other factors suggestive of influenza (predictive factors), to be used by the family doctor for the diagnosis of influenza, treatment and specific measures.
- evaluate the performance of different case definitions (WHO vs ECDC):
As a result, there are a lot of epidemiological data and information that make the article difficult to read and understand.
The introduction is very long, with a lot of general information about the influenza. I recommend to be restructured, keeping only the information relevant to the content of the article. The addition of national influenza surveillance data from the two seasons (2022/2023, 2023/2024) would help to better understand the characteristics of influenza in Serbia.
Authors’ response: Thank you for these comments. We have revised the Introduction section; however, due to the absence of national-level influenza surveillance data for the 2022/23 and 2023/24 seasons, we did not include this information in the Introduction. In this context, the aim of our study was not to analyze influenza-like illness or acute respiratory infection surveillance but rather to predict influenza among participants aged ≥15 years based on their characteristics, as presented in the Results section.
Reviewer's response. The introduction has been revised, I agree with the current form
The aim of the study is presented differently in the abstract, introduction, and discussions. It can be improved, redefined to reflect the results presented in the study and the same throughout the article.
Authors’ response: Thank you for these comments. We have revised the sentences outlining the study aim in the Abstract, Introduction, and Discussion sections according to your suggestions.
Reviewer's response. The aim of the study has been revised, I agree with the current form
Also the title can be improve.
Authors’ response: Thank you for this suggestion. However, we would like to emphasize that this study is part of an unpublished PhD thesis with the same title. To ensure that the article accurately reflects the process evaluation of the PhD thesis, we believe it is preferable to retain the current title of this paper.
Reviewer's response. I understand the explanation. I believe that the title of an article must reflect its content, even if this content is part of a doctoral thesis.
It is not explained why the population over 15 years of age was included in the study and not the entire population.
Authors’ response: Thank you for your consideration. As stated in the Introduction, previous research has evaluated the diagnostic validity of clinical features associated with influenza, with a stronger focus on pediatric populations than on adults. In adult populations, there is a paucity of data on how pre-existing health status, vaccination history, or comorbid conditions affect the diagnostic accuracy of case definitions.
Furthermore, as mentioned in the Introduction, only one prior study in our country has assessed the diagnostic accuracy of the WHO ILI case definition. This study included both ambulatory and hospitalized patients of all ages but did not consider pre-existing conditions, vaccination status, or contact history. Its findings demonstrated limited diagnostic utility, particularly in older adults, with low specificity and a high rate of false positives. Additionally, the study did not examine whether specific symptoms were more predictive of influenza type A or B or whether they varied by viral subtype [Ref. No 22: Ristić M, Petrović V. Evaluation of the diagnostic utility of case definitions to detect influenza virus infection in Vojvodina, Serbia. Srp Arh Celok Lek. 2020;148(1-2):100-05.].
Considering all of the above, our surveillance at the primary care level was conducted exclusively among general practitioners (GPs) rather than pediatricians. Consequently, we included only patients aged ≥15 years. Additionally, in the Materials and Methods section (Study Design), we have clarified that participant enrollment was conducted by general practitioners at the primary care level.
Reviewer's response. Thanks for the explanation.
I recommend that the reasons why the population over 15 years old was chosen (at the inclusion criteria) be explained more clearly in the text of the article, given that there are differences between countries in the definition of the adult population, etc
It is not clear how many family doctors participated in the study (all or those who wanted to?). Were there selection criteria?
Authors’ response: Thank you for this insightful question. The surveillance was designed to include five general practitioners (GPs) without specific selection criteria, except that these GPs had previously participated in influenza-like illness surveillance during past influenza seasons. However, only during the 2022/23 and 2023/24 seasons did they also collect patient specimens for laboratory testing for influenza. All of them received training on the inclusion and exclusion criteria for participant enrollment in this study.
Reviewer's response. I recommend that the selection criteria of the 5 family doctors be explained in the manuscript (e.g. 5 family doctors out of the total of n family doctors in the region wanted to participate in the study.....)
I recommend using the term influenza throughout the article (instead of flu).
Authors’ response: Thank you for this recommendation. We have replaced all instances of “flu” with “influenza” throughout the paper.
Reviewer's response. Thanks, the revision was made in accordance with the suggestion.
I recommend using the term “training” (instead of “education”), line 179.
Authors’ response: Thank you for this recommendation. We have replaced the term “education” with “training”.
Reviewer's response. Thanks, the revision was made in accordance with the suggestion.
Results
A lot of data, presented both in text and in tables.
Authors’ response: We fully agree with your comments. However, as mentioned above, this paper represents only a part of an unpublished PhD thesis, which includes more/additional results. Although our aim was to present only the most significant findings, a large amount of data remains in the paper. To maintain focus on the specific results, we have organized the paper so that other results are presented in the Supplementary section for readers who may be more interested. We believe that presenting our results in this manner will enable future researchers to review, compare, and evaluate our findings for their own studies.
Reviewer's response. Table 4 is very small, so the information could not be read.
223-224. A significant association was found between the absence of chronic diseases and testing positive for influenza (OR=3.30; 95% CI: 2.06–5.27; p<0.0001).
In my opinion this result should be explored and explained better, maybe they should assess this association by age group.
217-218. I cannot see the connection between timely administration of the COVID-19 vaccine and influenza. In addition, by timely administration of the COVID-19 vaccine I understand the administration of the vaccine immediately after the vaccine became available (probably in the first months of 2021), so without a connection with the seasons included in the study. However, if the authors consider important for the study results and conclusions to add this information, they should motivate.
The same comment for paragraph 466-471.
Line 223 and 274 - To be filled in with the number of patients. The number of samples is written in the text, but then the characteristics of the patients are presented.
Authors’ response: Thank you for these comments. We have corrected as “participant samples” instead of “samples”.
Reviewer's response. Thanks for the explanation. I suggest the following
187-190. Rephrase: A total number of 679 participants were included in the study, mostly female (442/679; 65.1%). Out of the total number of samples collected (679), 83 (12.2%) were tested positive for influenza. Various sociodemographic, clinical, epidemiological, and behavioral characteristics were significantly associated with laboratory-confirmed influenza (Table 1).
237-240. I propose a similar rephrasing as that proposed above for 187-190.
I propose to change “sore throat was significantly higher... “ with “sore throat was significantly associated with a negative result for influenza”, line 244.
Authors’ response: Thank you for this proposal. We have corrected the “Sore throat was significantly associated with a negative result for influenza”' instead of “Sore throat was significantly higher among influenza negative compared to influenza positive cases”.
Reviewer's response. Thanks, the revision was made in accordance with the suggestion.
Discussion
The discussion part is too long. I recommend keeping only what is relevant to the objectives and the results of this study.
Authors’ response: We fully agree with your comments once again. However, with your consent to retain the results of this study, we believe it is reasonable for the majority of the discussion to remain in its current form. As outlined in the Discussion section, we have focused solely on the statistically significant results of our study. Additionally, we have made an effort to explain why our findings differ from those of other relevant studies, while also highlighting similarities with the results of other researchers in this field. While we value the importance of your suggestion, we have streamlined the discussion to preserve the integrity of the key findings, avoiding unnecessary complexity in the interpretation. We believe that further shortening of the discussion could jeopardize the presentation of the main findings of our work.
Reviewer's response. I thank the authors for their efforts. In my opinion, the article is still difficult to understand.
Author Response
Response to Reviewver 1
Authors' response: We sincerely appreciate the thorough review of our manuscript and the valuable feedback provided. Your insightful comments and suggestions have significantly improved the clarity and quality of our work. We have carefully addressed all the points raised and revised the manuscript accordingly (please see the revised version and our detailed responses below).
The article “Diagnostic Significance of Symptoms and Signs of Influenza in a Population Aged ≥15 Years: Novi Sad, Serbia” presents the results of an observational prospective study on influenza carried out at primary care level in Novi Sad Serbia, during two influenza seasons (2022/23, 2023/24), in a population aged ≥15 Years.
This study presents the clinical factors associated with laboratory-confirmed influenza and the performance of influenza case definitions in patients aged ≥15 years, using data obtained from a primary healthcare influenza surveillance system.
The topic of research is relevant. The authors use surveillance data and advanced statistical tests; also the references are relevant. Nevertheless, the article requires major revision before accepting for publication.
Authors’ response: Thank you very much for the careful reading of the manuscript and your constructive suggestions. We have revised our manuscript according to your comments and suggestions point by point (please see the revised manuscript and following answers).
Comments and questions:
In my opinion, the presented study has 2 very ambitious objectives:
- to identify the signs, symptoms and other factors suggestive of influenza (predictive factors), to be used by the family doctor for the diagnosis of influenza, treatment and specific measures.
- evaluate the performance of different case definitions (WHO vs ECDC):
As a result, there are a lot of epidemiological data and information that make the article difficult to read and understand.
The introduction is very long, with a lot of general information about the influenza. I recommend to be restructured, keeping only the information relevant to the content of the article. The addition of national influenza surveillance data from the two seasons (2022/2023, 2023/2024) would help to better understand the characteristics of influenza in Serbia.
Authors’ response: Thank you for these comments. We have revised the Introduction section; however, due to the absence of national-level influenza surveillance data for the 2022/23 and 2023/24 seasons, we did not include this information in the Introduction. In this context, the aim of our study was not to analyze influenza-like illness or acute respiratory infection surveillance but rather to predict influenza among participants aged ≥15 years based on their characteristics, as presented in the Results section.
Reviewer's response. The introduction has been revised, I agree with the current form.
Authors’ response: Thank you for your agreement.
The aim of the study is presented differently in the abstract, introduction, and discussions. It can be improved, redefined to reflect the results presented in the study and the same throughout the article.
Authors’ response: Thank you for these comments. We have revised the sentences outlining the study aim in the Abstract, Introduction, and Discussion sections according to your suggestions.
Reviewer's response. The aim of the study has been revised, I agree with the current form.
Authors’ response: Thank you for your agreement.
Also the title can be improve.
Authors’ response: Thank you for this suggestion. However, we would like to emphasize that this study is part of an unpublished PhD thesis with the same title. To ensure that the article accurately reflects the process evaluation of the PhD thesis, we believe it is preferable to retain the current title of this paper.
Reviewer's response. I understand the explanation. I believe that the title of an article must reflect its content, even if this content is part of a doctoral thesis.
Authors’ response: Thank you for your comments. We have revised the title of our manuscript accordingly.
It is not explained why the population over 15 years of age was included in the study and not the entire population.
Authors’ response: Thank you for your consideration. As stated in the Introduction, previous research has evaluated the diagnostic validity of clinical features associated with influenza, with a stronger focus on pediatric populations than on adults. In adult populations, there is a paucity of data on how pre-existing health status, vaccination history, or comorbid conditions affect the diagnostic accuracy of case definitions.
Furthermore, as mentioned in the Introduction, only one prior study in our country has assessed the diagnostic accuracy of the WHO ILI case definition. This study included both ambulatory and hospitalized patients of all ages but did not consider pre-existing conditions, vaccination status, or contact history. Its findings demonstrated limited diagnostic utility, particularly in older adults, with low specificity and a high rate of false positives. Additionally, the study did not examine whether specific symptoms were more predictive of influenza type A or B or whether they varied by viral subtype [Ref. No 22: Ristić M, Petrović V. Evaluation of the diagnostic utility of case definitions to detect influenza virus infection in Vojvodina, Serbia. Srp Arh Celok Lek. 2020;148(1-2):100-05.].
Considering all of the above, our surveillance at the primary care level was conducted exclusively among general practitioners (GPs) rather than pediatricians. Consequently, we included only patients aged ≥15 years. Additionally, in the Materials and Methods section (Study Design), we have clarified that participant enrollment was conducted by general practitioners at the primary care level.
Reviewer's response. Thanks for the explanation.
I recommend that the reasons why the population over 15 years old was chosen (at the inclusion criteria) be explained more clearly in the text of the article, given that there are differences between countries in the definition of the adult population, etc
Authors’ response: Thank you for your comments. We have added an explanation on this topic in the Materials and Methods section (Study Design).
It is not clear how many family doctors participated in the study (all or those who wanted to?). Were there selection criteria?
Authors’ response: Thank you for this insightful question. The surveillance was designed to include five general practitioners (GPs) without specific selection criteria, except that these GPs had previously participated in influenza-like illness surveillance during past influenza seasons. However, only during the 2022/23 and 2023/24 seasons did they also collect patient specimens for laboratory testing for influenza. All of them received training on the inclusion and exclusion criteria for participant enrollment in this study.
Reviewer's response. I recommend that the selection criteria of the 5 family doctors be explained in the manuscript (e.g. 5 family doctors out of the total of n family doctors in the region wanted to participate in the study.....)
Authors’ response: Thank you for your comments. We have added an explanation on this topic in the Materials and Methods section (Study Design).
I recommend using the term influenza throughout the article (instead of flu).
Authors’ response: Thank you for this recommendation. We have replaced all instances of “flu” with “influenza” throughout the paper.
Reviewer's response. Thanks, the revision was made in accordance with the suggestion.
Authors’ response: Thank you for your agreement.
I recommend using the term “training” (instead of “education”), line 179.
Authors’ response: Thank you for this recommendation. We have replaced the term “education” with “training”.
Reviewer's response. Thanks, the revision was made in accordance with the suggestion.
Authors’ response: Thank you for your agreement.
Results
A lot of data, presented both in text and in tables.
Authors’ response: We fully agree with your comments. However, as mentioned above, this paper represents only a part of an unpublished PhD thesis, which includes more/additional results. Although our aim was to present only the most significant findings, a large amount of data remains in the paper. To maintain focus on the specific results, we have organized the paper so that other results are presented in the Supplementary section for readers who may be more interested. We believe that presenting our results in this manner will enable future researchers to review, compare, and evaluate our findings for their own studies.
Reviewer's response. Table 4 is very small, so the information could not be read.
Authors’ response: Thank you for your comment. We have adjusted the font size in Table 4 for better readability.
223-224. A significant association was found between the absence of chronic diseases and testing positive for influenza (OR=3.30; 95% CI: 2.06–5.27; p<0.0001).
In my opinion this result should be explored and explained better, maybe they should assess this association by age group.
Authors’ response: Thank you for this comment. The explanation likely lies in the fact that the mean age of participants with influenza was significantly lower (37.90 years) compared to those who tested negative (54.92 years) (p < 0.0001). Considering this, younger participants with influenza were more likely to be without comorbidities than to have them. We have also addressed this aspect in the Discussion section of the paper.
217-218. I cannot see the connection between timely administration of the COVID-19 vaccine and influenza. In addition, by timely administration of the COVID-19 vaccine I understand the administration of the vaccine immediately after the vaccine became available (probably in the first months of 2021), so without a connection with the seasons included in the study. However, if the authors consider important for the study results and conclusions to add this information, they should motivate.
Authors’ response: We agree with your comments. Participants who tested negative for influenza were more likely to have received timely COVID-19 vaccination. The observed negative association between influenza positivity and timely COVID-19 vaccination may reflect competing vaccination priorities or differences in health behavior patterns related to influenza and COVID-19, as stated in the Discussion section. In addition, it can be assumed that participants who were vaccinated against COVID-19 were also more likely to wear masks, maintain physical distance, and follow other preventive measures, which may have reduced the risk of influenza virus infection. We also added this sentence in the Discussion section.
The same comment for paragraph 466-471.
Authors’ response: We have revised the text in accordance with your suggestion.
Line 223 and 274 - To be filled in with the number of patients. The number of samples is written in the text, but then the characteristics of the patients are presented.
Authors’ response: Thank you for these comments. We have corrected as “participant samples” instead of “samples”.
Reviewer's response. Thanks for the explanation. I suggest the following
187-190. Rephrase: A total number of 679 participants were included in the study, mostly female (442/679; 65.1%). Out of the total number of samples collected (679), 83 (12.2%) were tested positive for influenza. Various sociodemographic, clinical, epidemiological, and behavioral characteristics were significantly associated with laboratory-confirmed influenza (Table 1).
237-240. I propose a similar rephrasing as that proposed above for 187-190.
Authors’ response: Thank you for this comments. We have revised the text in accordance with your suggestions.
I propose to change “sore throat was significantly higher... “ with “sore throat was significantly associated with a negative result for influenza”, line 244.
Authors’ response: Thank you for this proposal. We have corrected the “Sore throat was significantly associated with a negative result for influenza”' instead of “Sore throat was significantly higher among influenza negative compared to influenza positive cases”.
Reviewer's response. Thanks, the revision was made in accordance with the suggestion.
Authors’ response: Thank you for your agreement.
Discussion
The discussion part is too long. I recommend keeping only what is relevant to the objectives and the results of this study.
Authors’ response: We fully agree with your comments once again. However, with your consent to retain the results of this study, we believe it is reasonable for the majority of the discussion to remain in its current form. As outlined in the Discussion section, we have focused solely on the statistically significant results of our study. Additionally, we have made an effort to explain why our findings differ from those of other relevant studies, while also highlighting similarities with the results of other researchers in this field. While we value the importance of your suggestion, we have streamlined the discussion to preserve the integrity of the key findings, avoiding unnecessary complexity in the interpretation. We believe that further shortening of the discussion could jeopardize the presentation of the main findings of our work.
Reviewer's response. I thank the authors for their efforts. In my opinion, the article is still difficult to understand.
Authors’ response: Thank you for these comments. We greatly appreciate your opinion; however, we also emphasize the main strength of this study. As stated in the paper, the aim of our study was not to analyze influenza-like illness or acute respiratory infection surveillance but rather to predict influenza among participants aged ≥15 years based on their characteristics. Previous research has evaluated the diagnostic validity of clinical features associated with influenza, with a stronger focus on pediatric populations than on adults and more frequently among inpatients than at the primary care level. In adult outpatients, there is a paucity of data on how pre-existing health status, vaccination history, or comorbid conditions affect the diagnostic accuracy of case definitions. Except this, our research also showed that certain patient characteristics were useful for predicting influenza even when influenza prevalence was low.
Other studies were conducted mainly in the pre-pandemic COVID-19 period when influenza prevalence was higher. We highlighted these circumstances after reviewing 52 relevant references on this topic.
Given the extensive scope of the results, the discussion is necessarily lengthy; however, we believe it remains clear and precise.
Finally, we have made additional efforts to shorten the discussion to improve the clarity of the manuscript. We believe that further revisions of the existing text may lead to the omission of important result-related comments relevant to this study.
Reviewer 2 Report
Comments and Suggestions for Authors
The authors have addressed the reviewers' comment properly.
Author Response
Comments and Suggestions for Authors
The authors have addressed the reviewers' comment properly.
Authors’ response: Thank you for your agreement.
Round 3
Reviewer 1 Report
Comments and Suggestions for Authors
I thank the authors for their effort to improve the quality of the manuscript. I agree with the publication of the article in this updated and improved form.